psychology/behaviour/cognition

reciprocity, social influence, development, social norms, conformity

**Author for correspondence:**
Joshua Zonca
e-mail: joshua.zonca@iit.it

# I'm not a little kid anymore! Reciprocal social influence in child–adult interaction

Joshua Zonca[1], Anna Folsø[2] and Alessandra Sciutti[1]

[1]Cognitive Architecture for Collaborative Technologies (CONTACT) Unit, Italian Institute of Technology, Via Enrico Melen, 83, 16152 Genoa, Italy
[2]Department of Informatics, Bioengineering, Robotics and Systems Engineering, University of Genoa, Genoa, Italy

JZ, 0000-0002-6593-0968; AS, 0000-0002-1056-3398

Human decisions are often influenced by others' opinions. This process is regulated by social norms: for instance, we tend to reciprocate the consideration received from others, independently of their reliability as information sources. Nonetheless, no study to date has investigated whether and how reciprocity modulates social influence in child–adult interaction. We tested 6-, 8- and 10-year-old children in a novel joint perceptual task. A child and an adult experimenter made perceptual estimates and then took turns in making a final decision, choosing between their own and partner's response. We manipulated the final choices of the adult partner, who in one condition chose often the child's estimates, whereas in another condition tended to confirm her own response. Results reveal that 10-year-old children reciprocated the consideration received from the partner, increasing their level of conformity to the adult's judgements when the partner had shown high consideration towards them. At the same time, 10-year-old children employed more elaborate decision criteria in choosing when trusting the adult partner compared to younger children and did not show egocentric biases in their final decisions. Our results shed light on the development of the cognitive and normative mechanisms modulating reciprocal social influence in child–adult interaction.

## 1. Introduction

Humans often take advice and learn from others to optimize behaviour and decisions. This form of social influence minimizes the cost of individual learning [1–3] and solves uncertainty [4,5], promoting accumulation and transmission of knowledge across generations [6–8]. Several studies have shown that humans can effectively recognize when the information provided by peers is reliable and should be used to guide behaviour [9–12]. Moreover, they use information about informants' confidence [13,14] and

expertise [15–18] to choose the best sources of advice. Importantly, extensive evidence has shown that these abilities are present at an early age in human development. From around 3 to 4 years of age, children use individual characteristics of adults and peers, including expertise, confidence and age, to select the best informants for learning [19–23]. Moreover, children implement cognitive strategies to select the best informants and the most appropriate circumstances to learn and follow advice in social contexts [24–26].

Nonetheless, sometimes humans use suboptimal informational criteria for social learning and social decision making. One the one hand, this can happen due to cognitive biases: for instance, people tend to over-discount socially acquired information [27–30], overestimate their own abilities [31,32] and underestimate their own responsibility for undesired outcomes in cooperative settings [33,34]. On the other hand, distortions in the integration of social feedback can occur due to social norms: people tend indeed to conform to others to preserve self-image and reputation in their social group [35–38]. In this respect, several studies have shown that biases in judgement and decision making due to normative social influence emerge in early middle childhood [39–43]. Nevertheless, normative conformity tends to increase with age and achieves its peak in early middle adolescence [44–50]. A pervasive social norm regulating human behaviour is reciprocity: we tend to help who has helped us in the past and who could help us in the future. Reciprocity is considered as one of the key elements sustaining the emergence and the maintenance of cooperation between individuals and within social groups [51,52]. Furthermore, reciprocity is known as one of the most important factors sustaining the development of cooperative behaviour in childhood. Several studies have shown that children from 8 to 11 years of age reciprocate the trust received from a partner in the trust game [53–55]. However, reciprocal behaviour becomes more prominent during pre-adolescence and adolescence [53,56,57]. In this period of human development, reciprocity also becomes more elaborate [58]: for instance, adolescents' willingness to reciprocate is mediated by the risk taken by the partner (i.e. the investor) in the trust game [57,59]. Nonetheless, simpler forms of reciprocity can emerge earlier during childhood. Preschool-age children (5–6 years old) show reciprocity in anticipation to repeated interactions [60], match their own behaviour with the one of their counterpart during interaction [56] and reciprocate imitation received from an adult partner [61]. Moreover, preschoolers are able to anticipate future reciprocity of their interacting partners: they share more resources with partners who will have the possibility to reciprocate in the future [62,63] and expect to receive resources from whom have been favoured in the past [64]. Children as young as 4 years of age tend to distribute more resources to rich rather than poor recipients, if the former have shown an intention to reciprocate [65]. Three-year-old children selectively show pro-social behaviours towards pro-social partners [66,67], examine the intentions of partners to modulate pro-social behaviours towards them [68,69] and engage in costly punishment [70]. A preference for fair and generous donors has been found even in infants [71,72].

Interestingly, recent evidence [73] has shown that reciprocity can modulate social influence and advice taking in adult dyads: people tend to take more into account the opinion of someone who, in turn, shows high consideration towards their judgements. Another recent study [74] has shown that reciprocity of social influence emerges also indirectly in triadic relationships between adults. These results highlight that reciprocity regulates not only cooperative behaviour but also the exchange of information between adult individuals, which does not follow purely Bayesian principles of information aggregation [9]. Nonetheless, no study to date has investigated the developmental features of this phenomenon. In particular, we do not know whether and how children at different stages of development may reciprocate the consideration received from an adult. This aspect is fundamental in the understanding of the processes of establishment and consolidation of child–adult relationships, especially from an educational perspective, both in domestic (i.e. parent–child relationship) and scholastic (i.e. teacher–student relationship) environments. Children constantly receive an impressive amount of novel information from adults during development and sometimes these new pieces of information conflict with children's current knowledge. When children's beliefs and information from adults are in conflict, children may be unwilling to accept novel information and learn from adult informants [75]. In this context, children might claim a certain degree of consideration from adults in order to accept this new information, following the reciprocal mechanisms of social influence observed in dyadic and triadic interactions between adults [73,74]. In this sense, the emergence of social norms such as reciprocity might sustain child–adult interaction and collaboration, promoting children's learning by social interaction with adults.

In the current study, we investigate whether children from 6 to 10 years of age engage in reciprocal behaviour when making perceptual judgements and exchanging opinions about them with an adult partner. We selected this age range for several reasons. First, this age range is critical in the development of different forms of reciprocity, as already described in the literature review (e.g. [53,54]). In fact,

(*a*) experimental task

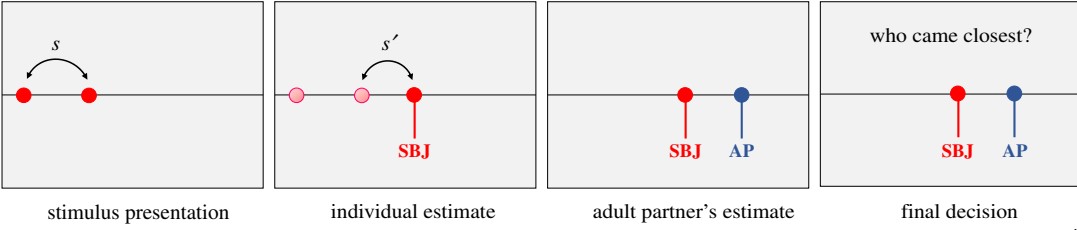

(*b*) experimental design

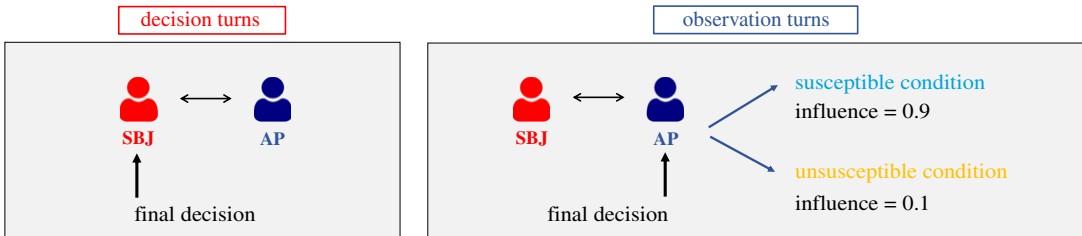

**Figure 1.** Experimental task and design. (*a*) Experimental task. At the beginning of each trial, participants saw two red discs appearing consecutively on a horizontal line. After their disappearance, participants had to touch a point, to the right of the second disc, in order to reproduce a segment of length (*s′*) that matched the target stimulus length (*s*, distance between the two red discs). At the same time, an adult partner (i.e. an experimenter) saw the same stimuli and simulated the reproduction of the stimulus length *s′* via mouse click. The estimates of both the participant and the adult partner were shown by red and blue disc underlined, respectively, by the real name of the child (here referred to as 'SBJ') and the real name of the adult partner (here referred to as 'AP'). Eventually, one of the two agents had the opportunity to make a final decision by choosing either her own or the other agent's response by touching the relative response point (or name) on the screen. Participants could see the final decisions of the adult partner and were aware that the latter could see the participant's final decisions. The final decision, for both agents, was highlighted by a green disc placed in the position of the selected response, which remained visible on the screen for 1 s. (*b*) Experimental design. In half of the trials (decision turns) the participant took the final decision, whereas in the other half of the trials (observation turns) the adult partner took the final decision. In the Susceptible condition, the adult chose the response of the child in 90% of the observation turns, whereas in the Unsusceptible condition the adult chose the child's response in 10% of the observation turns. In each condition, turns were arranged in blocks of five consecutive trials of the same type: first five observation turns, then five decision turns, and so on, for a total of 20 trials in each condition.

reciprocal social influence is a rather subtle and complex type of normative behaviour, in which individuals need to take into account several correlated factors, including informational uncertainty and confidence of the interactive partners, as well as the psychological and normative implications of giving and receiving consideration. For this reason, reciprocity in social influence contexts might emerge at an advanced stage of child development. Second, the selected age range is crucial in the emergence and evolution of normative mechanisms of conformity, which is known to become more and more prominent during childhood [39,40,41]. Third, children between 6 and 10 years of age live a crucial phase for the consolidation of child–adult relationships, both in scholastic and domestic educational settings. We highlight that a similar perceptual task has been successfully performed by children from 6 years of age in a previous work [76], confirming its suitability to our target population.

We introduce a novel interactive perceptual inference task (Reciprocal social influence task, figure 1). In each trial, a child and an adult partner made a perceptual estimate and then, in alternated experimental blocks, either the child or the adult took a final decision, choosing between own and other's judgement. We manipulated the adult partner's final decisions to create two different experimental conditions: Susceptible and Unsusceptible. In the former condition, the adult partner trusted with high probability the opinion of the child; in the latter condition, she tended not to consider the child's opinion and confirmed her own response. We compared children's behaviour across conditions to investigate the emergence of reciprocity of social influence at different ages.

We hypothesize that children may be more willing to accept the advice of the adult partner in the Susceptible condition, when the adult partner trusts the child's opinion, following reciprocal mechanisms. We expect this effect to be less pronounced, and possibly absent, in younger children

(e.g. 6-year-olds and perhaps 8-year-olds) due to the underdevelopment of social mechanisms linked to the understanding of social norms, as well as the cognitive abilities generally required in social interaction, including mentalizing, perspective taking and representation of the social decision context (see 'Research questions and statistical data analysis' for a detailed description of research questions and hypotheses).

# 2. Methods

## 2.1. Participants

We collected data from 65 children from an elementary school in Genoa (Italy). We tested children of three different classes of different educational levels: 1st grade ($n = 21$, 8 females, mean age ($M$) = 6 years and 4 months, standard deviation (s.d.) = 3.79 months), 3rd grade ($n = 21$, 12 females, $M = 8$ years and 4 months, s.d. = 5.13 months), 5th grade ($n = 23$, 13 females, $M = 10$ years and 3 months, s.d. = 3.09 months). The sample size was defined based on the demands and the availability of the involved elementary school and validated by statistical power analyses for sample size estimation. These power analyses focused on the predicted statistical analyses aimed at detecting a significant effect of age on reciprocity, following our main research question. The analyses for sample size estimation have been extensively described in the 'Sample size estimation' section, after a detailed description of the statistical analyses used along the manuscript in the 'Research questions and statistical data analysis' section.

The study was approved by the local ethics committee and the parents of all participants gave written informed consent prior to testing (see 'Ethical statement'). All participants had normal or corrected to normal vision acuity.

## 2.2. Reciprocal social influence task: experimental design, procedure and set-up

The experiment consisted of a partially simulated interaction between a child and an adult experimenter in the Reciprocal social influence task (figure 1). The entire data collection was carried out by the same adult experimenter, who is a woman of 43 years of age, psychotherapist and external collaborator of the Italian Institute of Technology. During school time, children took turns in following the experimenter in a neighbouring room set for the experimental session. Children were told that they would play a game with the experimenter. The experiment lasted approximately 25 min, including instructions and one break. During the experiment, the experimenter was instructed to maintain a neutral approach while interacting with children during the task. She did not provide either positive or negative comments concerning participants' behaviour and she did not comment on her own responses. In sum, the experimenter did not provide any physical or verbal feedback concerning the two agents' responses. The experimenter was asked to intervene verbally only if participants showed behaviour inconsistent with task instructions or were not focused on the task before an experimental trial. Moreover, during the experimental break, she was instructed to ensure that participants were able to continue the experiment.

Participants were seated in front of a touch-screen tablet ($27.67 \times 15.56$ cm) next to the experimenter (electronic supplementary material, figure S1, Experimental design and set-up). Each trial started with a double tap on the touch-screen from the participant to guarantee that the participant was focused on the task and ready to observe the visual stimuli appearing on the screen. At the beginning of each trial, participants had to reproduce the lengths of visual stimuli that consisted of two consecutive light flashes (red discs of 0.9 cm diameter, duration 500 ms) appearing on a visible horizontal white line crossing the screen at its central height. The first disc appeared at a variable distance from the left border of the screen (min: 0.8 cm, max: 3.5 cm, step: 0.3 cm). After the disappearance of the first disc and an inter-stimulus interval of 500 ms, the second disc appeared at a variable distance to the right of the first one. Participants were asked to reproduce the stimulus length $s$ (figure 1), defined as the distance between the first and the second disc, by touching a point on the white line, to the right of the second disc, with their index finger. In each trial, the distance $s$ between the two discs was randomly selected from 11 different sample distances (min: 4.3 cm, max: 9.7 cm, step: 0.6 cm). As soon as the children touched the screen, a third red disc marked with a vertical red line and the actual name of the children appeared in the selected position. We did not provide any feedback about the accuracy of the length estimation. Participants were also told that, during this interval, the adult partner would have made her response to estimate the length of the very same stimulus. The

experimenter pretended to make her length estimation using a computer mouse. We placed a partition panel between the child and the experimenter to prevent the participant from relying on the amplitude of the movement of the mouse of the experimenter to perform the task (electronic supplementary material, figure S1, Experimental design and set-up). Actually, we implemented a probabilistic algorithm to simulate the perceptual estimates of the adult partner. In each trial, the positions of the adult's perceptual estimate was randomly chosen from a Gaussian distribution centred at the correct response (s.d.: 1.06 cm). The standard deviation of the response error distribution was chosen to maintain a balance between variability, credibility and accuracy of response. Moreover, we wanted to always ensure a discrepancy between participant's and adult partner's responses ($d$). Therefore, whenever the sampled estimate of the algorithm was rather close to that of the participant (i.e. $d < 0.9$ cm), the algorithm re-sampled a new estimate from the distribution (i.e. until $d \geq 0.9$ cm). The choice of this set-up was necessary to have systematic control of the perceptual estimates of the adult partner to maintain comparability of the adult's performance across experimental conditions and groups.

After 1 s from the participant's response, the position of the adult partner's estimate, marked with a blue vertical line and the name of the experimenter, was shown. Participants knew that the adult partner was also able to see the position of both responses (the one of the children and the one of the adult) on screen. Right after the appearance of the adult's estimate, the final decision phase started. In this phase, one of the two agents (either the children or the adult) had the opportunity to make a final decision by selecting either their own or their partner's response. The two agents took the final decisions in different turns. We refer to *decision turns* as the trials in which *the children* had to make the final decision, while we refer to *observation turns* as the trials in which *the adult* took the final decision. Turns were arranged in blocks of five consecutive trials of the same type: first five observation turns, then five decision turns, and so on (electronic supplementary material, figure S2, Experimental design and set-up).

The experimental manipulation concerned the behaviour of the adult partner in terms of final decisions in the observation turns. In the Susceptible condition, the adult chose the response of the children in 90% of the cases, whereas in the Unsusceptible condition the adult chose the children's response in 10% of the trials. The sequence of the adult's choices was predetermined (electronic supplementary material, figure S2, Experimental design and set-up): in each condition, the inconsistent choice (i.e. selecting own response in the Susceptible condition, or vice-versa) was made during the first block of that condition. We counterbalanced the order of presentation of the Susceptible and the Unsusceptible conditions across participants and within age groups. In sum, participants performed a total of eight blocks (four observation and four decision turns), divided in two experimental conditions (two observation and two decision turns for each condition). In each condition and for all participants, the first block consisted of observation turns. After the first four blocks, we introduced a break to allow participants to rest. During the break, the experimenter was instructed to check whether participants were able to continue before re-starting the experiment.

Both the children and the adult made their final decisions by touching the screen in place of their own or the adult's response, or their respective names. At the time of making the final decision (decision turns), participants could see on the screen the sentence 'Who came closest? X (name of the participant) or Y (name of the adult partner)?'. During the observation turns, participants could see the sentence 'Now Y (name of the adult partner) will choose who came closest, in her opinion'. Both agents could see the final responses of the other agent. In particular, participants could see the final responses of the adult's partner and were aware that the latter could see their final decisions. The final decision of both agents was highlighted by a green disc appearing in the position of the selected perceptual estimate, which remained visible on the screen for 1 s.

The task was preceded by four training trials (two observation trials and two decision trials) to allow participants to familiarize with the task. In case participants did not show a full comprehension of the task, we provided additional training trials.

At the end of the task, participants were asked to evaluate from 1 to 10 their own and the adult partner's accuracy in terms of perceptual estimation.

## 2.3. Research questions and statistical data analysis

Our analysis approach aimed at testing two main research questions in the Reciprocal social influence task. First, we investigated how children at different stages of development are influenced by the feedback of the adult partner. In particular, we tested how children of different ages incorporate, represent and use the adult's feedback in their decision model. We have elaborated on this set of analyses in the *Social influence* section below. This set of analyses does not take into account the

manipulation implemented in the task, namely the type of experimental condition (Susceptible or Unsusceptible), which manipulates the susceptibility expressed by the adult partner towards children. Analysis of the effect of condition on children's final decisions aimed at testing our main research question, targeting the emergence of reciprocity of social influence in children of different ages. We refer to this second cluster of analyses in the *Reciprocity* section.

### 2.3.1. Social influence

We analysed the general children's tendency to follow the adult partner in their final decisions as a function of their age and a set of endogenous variables linked to the informational context of the decision (i.e. estimation error, distance from the partner's estimate, performance ratings). The age variable was treated both as a continuous variable (months of age/12) and a group factor (6-, 8- and 10-year-olds) in distinct sets of analyses. The *estimation error* expresses the distance from the child's estimate to the correct response (in cm), divided by the current stimulus length (in cm) to give equal weight to trials including short and long visual stimuli. The same normalization procedure has been applied to the distance (in cm) between child's and adult's estimates (*agents' response distance*). Performance ratings were analysed by computing the relative difference between rating of own and other's accuracy, to adjust for individual differences in terms of reference point within the rating scale (1–10). We explored the contribution of these endogenous variables in modulating children's final decisions using multiple regressions and correlation analyses on individual variables (e.g. continuous age, mean estimation error, mean agent's response distance) as well as mixed-effects logistic models on trial-by-trial variables. In the former case, final decisions were characterized computing a main dependent variable called *influence*, defined as the proportion of trials in which participants chose their partner's estimate in their final decisions. In the latter case, final decisions were analysed as binary dependent variables (1 = own estimate chosen in the final decision, 0 = partner's estimate chosen in the final decision). In mixed-effect models, the random effects were applied to the intercept at the subject level, to adjust for the baseline level of influence of each subject and model intra-subject correlation of repeated measurements. The variance–covariance matrix of all regressions and models was estimated using robust variance estimator [77–79] to obtain heteroscedasticity-robust standard errors clustered at the subject level. All models analysing trial-by-trial final decisions were also run adjusting for trial-by-trial potential confounding variables such as estimation error and agents' response distance. These control analyses were run separately since participants' estimation error and agents' response distance are highly correlated. Specification and results of each model have been described in detail in the electronic supplementary material ('Data analysis and results' paragraph). In the 'Results' section of the main text, we report unstandardized regression coefficient (*B*), *t* statistic (*t*) and *p*-value for mixed-effect models and regression analyses. In the electronic supplementary material, we report complete fixed-effects results including robust standard errors and confidence intervals. In the case of regressions with multiple continuous independent variables, we also report standardized regression coefficients (*β*) for between-predictor comparison.

Moreover, we compared endogenous variables such as estimation error, agents' response distance and performance ratings across age groups to characterize the current informational context and the relative participant's representation. Since these individual variables occasionally show some degree of skewness and, in some cases, show a violation of the normality distribution assumption, we used non-parametric tests (i.e. Wilcoxon signed-rank test; Wilcoxon rank-sum test) through the entire paper for consistency. For the same reason, we used non-parametric correlation tests (Spearman's rank correlation). All tests are two-tailed and report *z* statistic, *p*-value and effect sizes ($r$, $\eta^2$). The formulae used for the calculation of the effect sizes can be found in Cohen [80] and Fritz *et al.* [81]: $r = Z/\sqrt{N}$ (total number of observations); $\eta^2 = Z^2/N$; $d$ (used in the power analysis for sample size estimation) $= 2r/\sqrt{(1 - r^2)}$.

### 2.3.2. Reciprocity

This set of analyses aims at investigating if children's final decisions were affected by the experimental condition (Susceptible or Unsusceptible). In particular, we tested whether children's willingness to follow the adult partner's opinion was higher in the Susceptible condition, following reciprocal mechanisms rather than Bayesian principles of information aggregation [9]. This effect has been explored as a function on participants' age, again treated both as continuous and categorical variable. The two analyses aimed at exploring two different facets of our main experimental hypothesis: the former aimed at testing the existence of a linear increase in the occurrence of reciprocal behaviour

along with age, the latter aimed at investigating the emergence of reciprocal behaviour at a specific step of child development. When age was treated as continuous variable, we used multiple regression analysis with age as independent variable and *reciprocity* as dependent one, adjusting for potential confound variables such as estimation error, mean individual level of social influence and performance ratings. *Reciprocity* was computed as the difference between *influence* (i.e. proportion of trials in which participants chose their partner's estimate in their final decisions) in Susceptible and Unsusceptible conditions for each participant. When treating age as group factor, we used mixed-effects model with trial-by-trial (binary) final decisions as dependent variable and age, condition and their interactions as independent factors. Random effects and standard errors were treated as in the previous cluster of analyses. As in the previous cluster of analyses, the main model was run also controlling for trial-by-trial potential confounding variables such as estimation error and agents' response distance. To corroborate the findings of the mixed-effects models, we also compared influence across conditions in each age group, using two-tailed Wilcoxon signed-rank tests with Bonferroni correction. We reported statistics and effect sizes using the same procedures described in the previous paragraph (Social influence).

## 2.4. Sample size estimation

Participants in our sample were recruited from an elementary school in Genoa (Italy). The school offered the possibility to conduct the experiment on three different classes of educational levels (1st, 3rd and 5th grade). All three classes included around 20–25 children. After a preliminary discussion with the children's parents and teachers, we expected a large part of the sample pool to participate in the experiment. The expected sample size was in line with the typical sample sizes in the field. Nonetheless, we also ran statistical analyses for sample size estimation to confirm the validity of our sample size. The target sample size was estimated based on the statistical analyses aimed at detecting a significant effect of age on reciprocity, treating age both as a continuous and a categorical variable. The former analysis investigates the existence of a linear increase in the occurrence of reciprocal behaviour along with age, whereas the latter aims at exploring the emergence of reciprocal behaviour at a specific developmental stage. These analyses have been extensively described in the 'Reciprocity' paragraph in the 'Research questions and statistical data analysis' section.

Concerning the former analysis (age treated as continuous variable), we planned to run a multiple regression analysis with *reciprocity* as the dependent variable and age as continuous independent variable, adjusting for the potential confound variables described in the previous section. Importantly, to our knowledge, there are no previous results showing a between-subject effect of age on reciprocal social influence and therefore we did not have any priors about the magnitude of the hypothesized effects. We have, therefore, assumed a classical medium effect size ($f^2 = 0.15$) along with a two-tailed distribution and a power of 0.8 to obtain the required sample size for a multiple linear regression analysis. The required total sample size resulting from this analysis is 55, which is in line with our expected sample.

Concerning the second type of analysis (effect of condition within age groups), we characterized the expected effect size for the sample size estimation by computing the minimum *within-subject* effect of condition observable in our dependent variable (*influence*). Given that children face 10 decision trials in each condition, and influence is calculated as the proportion of trials in which participants chose their partner's estimate in their final decisions, the minimum within-subject difference we can expect is 0.1, consistent with a difference of one choice across conditions. Assuming an average standard deviation of 0.15 in the difference in influence across conditions, we can calculate the relative effect size, which is 0.67. This effect size is in line with the assumption of a medium ($d = 0.5$) to large ($d = 0.8$) effect size. Actually, the estimated effect size is relatively conservative if compared to a recent study [73] investigating the emergence of reciprocal social influence in adults. In two experiments, the authors have shown that the adults' willingness to follow the opinion of an (alleged) adult partner was higher when the partner herself had shown high consideration (Susceptible condition) rather than low consideration (Unsusceptible condition) towards them. The effect size of the observed within-subject effect of condition (Susceptible–Unsusceptible) was considerably large in both experiments (Exp. 1: $r = 0.57$, $d = 1.39$; Exp. 2: $r = 0.46$, $d = 1.03$).

Therefore, we calculated the expected sample size for a Wilcoxon signed-rank test testing the effect of condition on *influence* in each age cohort, assuming an effect size of 0.67, a power of 0.8 and a two-tailed distribution. The analysis returns a sample size of 21 for each age group, which is in line with the requirements of the previous analysis (i.e. a total of 55 participants across age bands) and the demands of the elementary school in which the study was conducted.

## 2.5. Exclusion of participants

We aimed to exclude participants who did not comply with the instructions or were not able to remain focused on the task. We, therefore, analysed participants' *inconsistent responses*, which are defined as perceptual inferences spatially located to the *left*, rather than to the *right*, of the second of the two discs constituting the reference visual stimulus in the perceptual task. These responses, independently of participants' perceptual accuracy, are by definition incoherent with task instructions, which invite participants to touch a point to the right of the second disc. We calculated, for each participant, the proportion of inconsistent responses over the total number of trials (40), and we excluded participants whose proportion was two standard deviations above the sample mean (sample mean = 0.03, s.d. = 0.09). We, therefore, excluded three participants (6 y/o) whose percentage of inconsistent responses was higher than the 21% of the total number of trials (i.e. more than eight trials). These participants, respectively, showed a percentage of 22.5%, 37.5% and 60% of inconsistent responses. The remaining sample ($n = 62$) showed a very low proportion of inconsistent responses (mean = 0.01, s.d. = 0.03), with a maximum of 15% of inconsistent responses (six trials) in one participant. Seventy-seven percent of participants did not exhibit any inconsistent responses, while 90% of participants showed a maximum of *one* inconsistent response. In sum, all the analyses reported in the current paper include 62 participants (6 y/o, $n = 18$; 8 y/o, $n = 21$; 10 y/o, $n = 23$).

# 3. Results

## 3.1. Social influence

First, we analysed developmental differences in the amount of social influence exerted by the adult partner on children in the Reciprocal social influence task, independently of the experimental condition. We show a significant relationship between participants' *influence*, computed as the proportion of trials in which the child chose the adult's response in her final decisions, and age (linear regression with robust standard errors, standardized coefficient ($\beta$) = 0.52, unstandardized coefficient ($B$) = 0.06, $t = 4.79$, $p < 0.001$). This result does not change when adjusting for agents' response distance (effect of age: $\beta = 0.52$, $B = 0.06$, $t = 3.80$, $p < 0.001$; effect of distance: $\beta = -0.01$, $B = -0.01$, $t = -0.04$, $p = 0.967$) or participants' estimation error (effect of age: $\beta = 0.52$, $B = 0.06$, $t = 3.64$, $p = 0.001$; effect of error: $\beta = -0.01$, $B = -0.01$, $t = -0.05$, $p = 0.964$).

To better characterize the developmental features of social influence, we ran a mixed-effect logistic regression with participant's final decision (self/other) as the dependent variable, age (6, 8 and 10 y/o) as categorical factor and subject as random effect (electronic supplementary material, Data analysis and results, model 1). Results (figure 2a) show that 10-year-old children were more influenced by the adult partner than 6-year-old (6 y/o–10 y/o: $B = 1.07$, $z = 4.56$, $p < 0.001$) and 8-year-old children (8 y/o–10 y/o: $B = 0.64$, $z = 3.09$, $p = 0.002$). The difference between 8- and 6-year-old children is only marginally significant (6 y/o–8 y/o: $B = 0.44$, $z = 1.70$, $p = 0.090$). We obtain the same effects when adjusting for the distance between the two agents' responses and participants' estimation error in the previous model (electronic supplementary material, Data analysis and results, models 2, 3). In this respect, participants' estimation error in all the three age groups was significantly higher than the one of the adult partner (i.e. computer algorithm) (Wilcoxon signed-rank test. 6 y/o: $z = -5.12$, $r = 0.85$, $\eta^2 = 0.73$, $p < 0.001$; 8 y/o: $z = -3.91$, $r = 0.60$, $\eta^2 = 0.36$, $p < 0.001$; 10 y/o: $z = -4.29$, $r = 0.63$, $\eta^2 = 0.40$, $p < 0.001$. All results are significant at the Bonferroni corrected threshold for three comparisons). We also acknowledge that participant's average estimation error was higher in 6-year-old children than in the other two age groups (6 y/o: mean ± s.d. = 0.62 ± 0.29; 8 y/o: mean ± s.d. = 0.26 ± 0.13; 10 y/o: mean ± s.d. = 0.22 ± 0.06). Regression results show that the differences between 6-year-old children and the other groups are statistically significant (8 y/o–6 y/o: $B = -0.29$, $t = -4.42$, $p < 0.001$; 10 y/o–6 y/o: $B = 0.33$, $t = -5.49$, $p < 0.001$), whereas there is no difference between 8- and 10-year-old participants in terms of estimation error (10 y/o–8 y/o: $B = -0.042$, $t = -1.32$, $p = 0.192$). However, we did not find any relationship between across-subject variability in individual mean estimation error and between-subject heterogeneity in influence in none of the three age groups. (6 y/o: Spearman correlation, $\rho = -0.04$, $p = 0.859$; 8 y/o: $\rho = 0.19$, $p = 0.409$; 10 y/o: $\rho = -0.14$, $p = 0.522$). This result suggests that, at every age level, the general participant's susceptibility to the partner's opinion was not significantly explained by the individual perceptual error. This is not surprising since participants did not receive any feedback about their own and their partner's accuracy along the experiment.

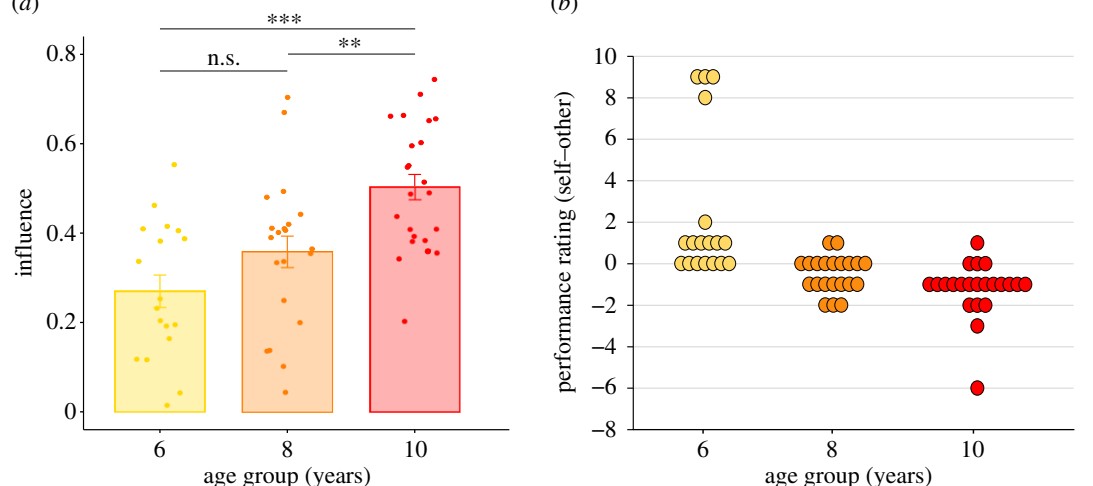

**Figure 2.** Effect of social influence across age groups. (*a*) Influence exerted by the adult partner on children of different ages (6-, 8-, 10-year-olds). Influence is defined as the proportion of decision trials in which the participant chose the estimate of the adult partner in her final decision. Error bars represent between subject standard errors of the mean. **$p < 0.01$, ***$p < 0.001$, n.s.: not significant. Effect of age group in Mixed-effect model. (*b*) Participants' performance rating (own accuracy rating−adult partner's accuracy rating) across age groups.

Furthermore, we controlled for potential age-dependent learning effects along the experiment. We divided the experimental task in 4 blocks of 10 trials each to investigate the temporal evolution of participants' estimation error. Results of a mixed-effect model (electronic supplementary material, Data analysis and results, model 4) with estimation error as the dependent variable, block, age group and their interaction as independent factors did not reveal any effect of block or interaction between age and block (contrasts of marginal linear predictions, block: d.f. = 3, $\chi^2 = 1.44$, $p = 0.696$; group × block: d.f. = 6, $\chi^2 = 3.09$, $p = 0.797$).

Results of performance ratings (figure 2*b*) confirmed than 6-year-olds believed to be more accurate than their adult partner (own rating–partner rating: mean + s.d. = 2.47 ± 3.64; median = 1. Wilcoxon signed-rank test, $z = 3.09$, $r = 0.53$, $\eta^2 = 0.28$, $p = 0.002$) while 8-year-olds and 10-year-olds showed the opposite pattern (8 y/o:−0.52 ± 0.87; median = 0. Wilcoxon signed-rank test, $z = -2.39$, $r = 0.37$, $\eta^2 = 0.14$, $p = 0.017$; 10 y/o: −1.23 ± 1.34; median = −1. Wilcoxon signed-rank test, $z = -3.79$, $r = 0.57$, $\eta^2 = 0.33$, $p < 0.001$). Results of 6- and 10-year-old children are significant at the Bonferroni-corrected threshold (*n*. comparisons = 3), while the effect of 8-year-olds is slightly above the border of Bonferroni-corrected significance threshold (Bonferroni-corrected $p = 0.051$). Regression results (own rating–partner's rating as dependent variable, age group as dummy factor, robust standard errors) confirm that the differences between the ratings of 6-year-old children and the other two participant cohorts are statistically significant (6 y/o–8 y/o: $B = 2.99$, $t = 3.33$, $p = 0.002$; 6 y/o–10 y/o: $B = 3.70$, $t = 4.00$, $p < 0.001$).

Furthermore, we tested whether children of different age groups modulated trial-by-trial final choices as a function of the current distance between the two agents' responses, which is the only feedback available to participants for extracting information about the relative difference between the two agents' performance. Results of a mixed-effects model (electronic supplementary material, Data analysis and results, model 5) show that 10-year-old children modulated their susceptibility towards the adult partner in function of the current agents' response distance ($B = 0.99$, $z = 2.17$, $p = 0.030$), with higher distances leading to a lower probability to follow the partner's advice. Conversely, younger children did not show any relationship between distance and influence (6 y/o: $B = 0.04$, $z = 0.17$, $p = 0.863$; 8 y/o: $B = 0.27$, $z = 0.73$, $p = 0.464$), suggesting that they did not take into account the relative difference in the accuracy of two agents to take the final decision.

## 3.2. Reciprocity

Then we analysed the developmental trajectory of reciprocal social influence. In particular, we tested whether the individual tendency to accept the advice of the adult partner depended on the

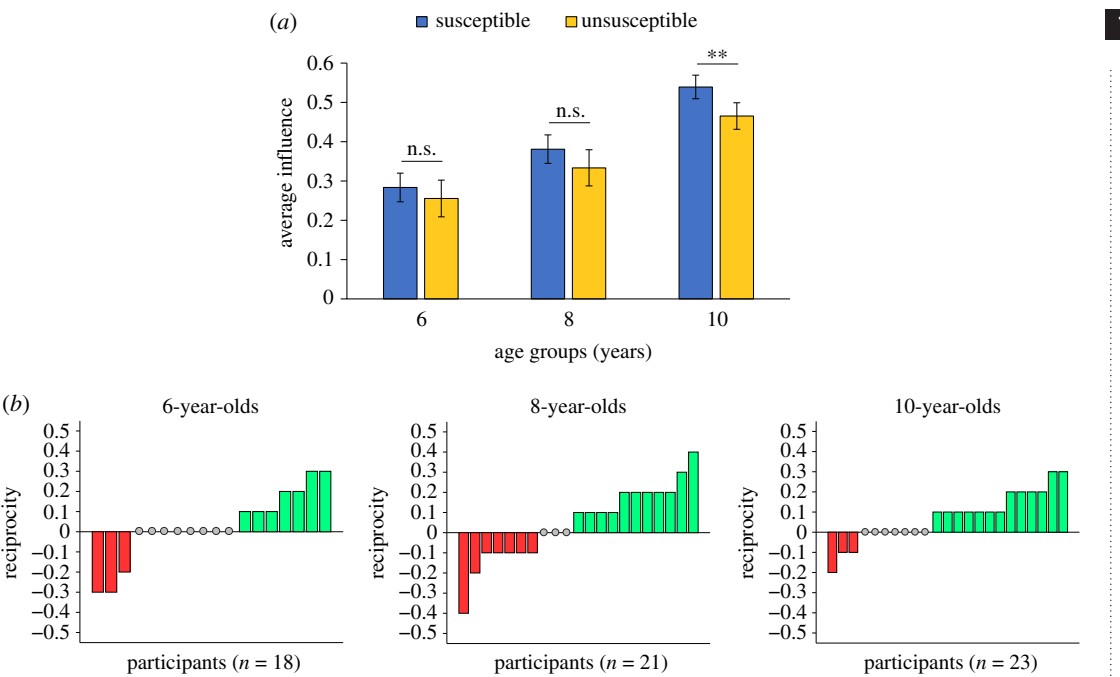

**Figure 3.** Effect of reciprocity across age groups. (a) Average influence in Susceptible and Unsusceptible conditions across age groups (6-, 8-, 10-olds). $^{**}p < 0.01$, ns: not significant. Effect of the condition by age group in Mixed-effect model. (b) Reciprocity index, calculated as influence in the Susceptible condition minus influence in the Insusceptible condition, plotted across participants and divided for age group. Positive values (in green) represent children who showed reciprocal behaviour, choosing more often the adult's partner estimate in the decision trials of the Susceptible condition, compared to the Unsusceptible one. Negative values (in red) represent participants who showed anti-reciprocal behaviour and selected less often the partner's estimate in the Susceptible condition. Grey dots represent individuals whose influence was identical across conditions.

susceptibility of the adult herself towards the children in the two experimental conditions (Susceptible or Unsusceptible). First, we found that reciprocity, defined as the difference between participants' influence in the Susceptible and the Unsusceptible condition, is significantly predicted by age (multiple linear regression with robust standard errors, $\beta = 0.43$, $B = 0.04$, $t = 2.25$, $p = 0.029$), while it is not explained by potential confound variables such as participants' estimation error ($\beta = 0.16$, $B = 0.12$, $t = 1.44$, $p = 0.157$), average influence ($\beta = -0.27$, $B = -0.24$, $t = -1.73$, $p = 0.089$) and performance ratings ($\beta = 0.13$, $B = 0.01$, $t = 0.81$, $p = 0.424$).

In order to test whether reciprocal social influence emerges at a specific stage of child development, we also ran a mixed-effect logistic regression (electronic supplementary material, Data analysis and results, model 6) with participants' final decision as dependent variable, condition (Susceptible or Unsusceptible), age group (6, 8 and 10 y/o) and their interactions as independent factors, and subject as random effect. Results show a significant effect of condition in 10-year-old children ($B = 0.31$, $z = 2.86$, $p = 0.004$), who switched more opinion towards the one of the adult partner when the latter has previously shown high consideration towards them (figure 3a,b). This effect was absent in younger children (6 y/o: $B = 0.15$, $z = 0.69$, $p = 0.489$; 8 y/o: $B = 0.22$, $z = 1.18$, $p = 0.239$). These findings were confirmed when controlling for agents' response distance and participants' estimation error (electronic supplementary material, Data analysis and results, models 7, 8). In this respect, we highlight that agents' response distance in decision turns was comparable across conditions in all the three age groups (Wilcoxon signed-rank test, 6 y/o: $z = 0.41$, $r = 0.07$, $\eta^2 = 0.00$, $p = 0.679$; 8 y/o: $z = -1.23$, $r = 0.19$, $\eta^2 = 0.04$, $p = 0.217$; 10 y/o: $z = -0.33$, $r = 0.05$, $\eta^2 = 0.00$, $p = 0.738$). Similarly, we did not find any between-condition difference in terms of participants' estimation error in any of the three age groups (Wilcoxon signed-rank test, 6 y/o: $z = 0.33$, $r = 0.05$, $\eta^2 = 0.00$, $p = 0.744$; 8 y/o: $z = -1.75$, $r = 0.27$, $\eta^2 = 0.07$, $p = 0.079$; 10 y/o: $z = -0.33$, $r = 0.05$, $\eta^2 = 0.00$, $p = 0.903$).

Our main effect of reciprocity were corroborated by Wilcoxon signed-rank tests showing that influence was significantly higher in the Susceptible than in the Unsusceptible condition in 10-year-old children ($z = 2.53$, $r = 0.37$, $\eta^2 = 0.14$, $p = 0.011$, significant at Bonferroni-corrected threshold, number of comparisons = 3). This effect is absent in 8-year-old ($z = 1.29$, $r = 0.20$, $\eta^2 = 0.04$, $p = 0.198$) and

6-year-old children ($z = 0.99$, $r = 0.16$, $\eta^2 = 0.03$, $p = 0.323$). Altogether, these results reveal that 10-year-old children engage in reciprocal behaviour in our social influence scenario.

## 4. Discussion

Recent evidence [73] has shown that humans engage in reciprocal behaviour when exchanging opinions with a peer. In particular, humans take more into account a partner's opinion when the partner has previously shown high consideration towards them, independently of her reliability as informant. However, no study to date investigated the developmental features of reciprocal social influence in child–adult interaction. We designed a novel experimental paradigm to investigate the emergence of this phenomenon in 6-, 8- and 10-year-old children. Participants performed an interactive perceptual task with an adult experimenter. Both agents produced a perceptual estimate of a common visual stimulus, and then took turns making a binary final decision, either choosing to confirm their own response or selecting the partner's estimate. We manipulated the final decisions of the adult partner implementing two different experimental conditions: in the Susceptible condition, the adult partner very often chose the child's estimate, whereas in the Unsusceptible condition she tended to confirm her own estimate. We compared participants' final decisions in the two conditions to explore the emergence of reciprocity in children of different ages. Results indicate that the emergence of reciprocal social influence depends on participants' age. In particular, 10-year-old children did reciprocate the consideration received from the adult partner, increasing their willingness to choose the partner's opinion in the Susceptible condition. This result held when controlling for informational factors associated with the perception of the two agents' abilities, such as estimation accuracy, agents' response distance and performance ratings. The emergence of reciprocal behaviour is inconsistent with Bayesian principles of information aggregation [9,73,74,82], which would predict higher participants' susceptibility during an interaction with an unsusceptible partner, since, from an informational point of view, susceptibility can be interpreted as a signal of behavioural uncertainty or incompetence. For this reason, reciprocal behaviour in social influence contexts is generally interpreted as an inherently normative mechanism sustained by the desire to maintain influence over others and avoid the distress of being ignored [73,74,83].

By contrast, younger children (6- and 8-year-olds) did not change their final decisions as a function of the partner's behaviour. We highlight that these children also showed a lower general susceptibility towards the adult partner's opinion compared to 10-year-old participants. On average, 10-year-olds chose the partner's option in about 50% of the trials, in line with previous results involving pre-adolescents and adolescents [58]. Conversely, children of 6 and 8 years of age exhibited a considerable egocentric advice discounting [29,30] and selected the partner's estimate in only 27% and 36% of the trials, respectively. These findings are consistent with extensive evidence suggesting that normative conformity increases during development, achieving its peak in early middle adolescence [41–50]. Furthermore, 10-year-old participants implemented more sophisticated decision strategies and criteria in their final choices. In particular, they took into account the contingent perceived reliability of the partner's estimate, modulating the probability of accepting their partner's advice depending on the distance between the two perceptual estimates. On the contrary, younger children did not modulate their susceptibility towards the adult partner based on the perceived reliability of her response, maintaining a more consistent pattern of decisions that favoured the confirmation of their own responses and refused the partner's advice. This effect is in line with theories of metacognition [84–86] arguing that preschool children select informational sources for learning by domain-general psychological processes and relatively simpler heuristics, whereas the ability to use domain-specific, rule-based mechanisms to guide selective social learning arises at later stages of development. Following this interpretation, in our task 10-year-old children may be better at evaluating, representing and updating the relevant informational context along the experiment, adapting their decision strategy based on the trial-by-trial perceived performances of the two interacting agents.

The marked egocentric bias observed in 6- and 8-year children is in line with evidence showing that children do not trust adults indiscriminately, but rely on epistemic considerations while judging their reliability as informants [75]. Infants as young as 16 months react with surprise when an adult labels objects incorrectly and then try to correct the speaker [87]. Toddlers around 2 years of age trust more speakers that have previously labelled objects correctly, revealing the emergence of selective trust in early child–adult interaction [75]. Furthermore, children between 3 and 4 years of age are sensitive to the relevant expertise of adults [88] and use this information to evaluate their testimony [89]. In this

regard, we acknowledge that, in our task, the tendency of 6- and 8-year-olds to refuse the advice of the adult partner was detrimental for task performance, since the adult partner's accuracy was markedly higher than that of children at every age. This poor assessment of self and other's performances may be linked to the underdevelopment of the relevant metacognitive abilities. In fact, we know that metacognition improves with age, reaching its peak in late adolescence [90]. The development of metacognitive abilities is fundamental for the refinement of social skills and can be reinforced, in turn, by social and cultural learning [91].

In sum, we highlight three main findings: (i) 10-year-old children reciprocated the consideration received from the partner, increasing their level of susceptibility when the partner had shown high consideration towards them; (ii) 10-year-old children were generally more influenced by the opinions of an adult partner, compared to 8- and 6-year-old children; (iii) 10-year-old children were more strategic in selecting the circumstances under which they followed their partner's advice. Taken together, our results suggest that, from the age of 10, children engage in a richer type of interaction with adults, both from informational and normative points of view. An interesting question is whether the ability of implementing sophisticated normative and informational behaviour during interaction emerges independently during development or if, alternatively, the application of reciprocation mechanisms is a direct consequence of a more sophisticated type of social interaction. In line with the latter view, several studies have suggested that the implementation of reciprocal behaviour in cooperative settings follows the development of specific cognitive mechanisms necessary for engaging in sophisticated interactions. One of these cognitive mechanisms is perspective taking: children reciprocate only when they are able to take the perspective of their counterpart and process and understand their goals and intentions. These abilities are already present at 4 years of age [92], but develop and become more and more sophisticated during adolescence, reaching their peak in 15- and 16-year-olds [57,59,93]. Furthermore, reciprocity requires a correct and exhaustive representation of the individual goals, incentives and potential actions of the interacting partners, as well as a full comprehension of the impact of one's choices on the counterpart. In this respect, extensive evidence in social decision making and behavioural economics has shown that these processes require specific cognitive abilities, including first- and second-order mentalizing and cognitive reflection [94–99]. These abilities emerge in preschool age [100] but develop and refine along childhood and early adolescence, around 6 and 12 years of age [101–103]. Moreover, in order to engage in reciprocal behaviour, children should override selfish impulses in view of a long-term benefit in terms of affiliation and cooperation with others, an ability that emerges between 3 and 5 years of age and improves during childhood [104–106]. In sum, being able to form a sophisticated representation of the interactive scenario may be necessary to engage in reciprocation in social influence settings. In our experimental task, children need indeed to construct a complex representation of the behaviour of the current partner: they must integrate individual motives concerning the attempt to maximize accuracy and, at the same time, the willingness to establish an affiliation with the interacting partners. This interpretation is consistent with the result showing that elaborate forms of reciprocity, which take into account the costs and the benefits associated with reciprocal behaviour, emerge only at late stages of childhood [53,54] or during pre-adolescence and adolescence, reaching its peak at 15 and 16 years of age [57,59,93]. We hypothesize that 10-year-old children developed the minimal cognitive requirements to interpret the adult's behaviour in our task, combine and balance considerations about general performance, contingent accuracy and social norms, and implement a hybrid and dynamic decision model that lead to either conforming or egocentric behaviour depending on present and past informational and normative contexts.

Furthermore, in social influence contexts, the emergence of reciprocity requires the possession of specific knowledge about the social norms that regulate advice taking in everyday life. In particular, children should acknowledge that their partner's susceptibility might be influenced by a desire to please them and trust them, and not only by uncertainty or incompetence. It is possible that knowledge about the normative mechanisms modulating advice taking and mutual trust consolidates as a social norm at a late stage of child development, unlike simpler forms of normative behaviour (e.g. helping others). We believe the contextual development of both cognitive and social abilities to play a key role in the emergence of reciprocal social influence in childhood.

Eventually, we should acknowledge that our findings shed light on the emergence of reciprocal social influence in a particular type of social interaction, specifically the one occurring between a child and an adult. Child interactions with peers may be regulated by different mechanisms. In fact, we know that conformity in children and adolescents depends on the age of the partners [41,58]. Moreover, the type of social relationship that children establish with adults is inherently different from the one that they

establish with other children. In particular, reciprocal behaviour may emerge more easily in children who play or collaborate with each other on a regular basis (e.g. friends), rather than in child–adult interactions in which the adult assumes a peculiar educational or authoritarian role (e.g. teacher, parent). Further research should explore the specific mechanisms regulating the interplay between reciprocity and social influence in different types of child–child and child–adult interactions. Moreover, we acknowledge that the social scenario implemented in the current study entails an interaction with an adult partner then changes their susceptibility to the child's estimates along the task, following a within-subject design. This type of scenario is particularly interesting since it can reveal the dynamic nature of reciprocity, which typically regulates behavioural adaptation in dynamic and unstable social environments. Moreover, it can provide interesting insights about children' sensitivity to adults' behavioural switches concerning social norms and their adaptive reactions to such norm violations. Nonetheless, this is just one of the several ways in which susceptibility can be manipulated. For instance, future studies may investigate children's behavioural reactions to the behaviours of two adults showing different levels of susceptibility: in this context, the adult partners' behaviour might be easier to represent and characterize by children, which may facilitate the emergence of reciprocal behaviour.

Taken together, our results highlight an important component of social influence in childhood, suggesting that children of 10 years of age can engage in reciprocal behaviour when exchanging opinions with an adult partner. At this age, children start to react to the consideration they receive from adults and are more willing to take into account the opinion of someone who listens to, rather than ignores, them. We believe these findings to have important implications at the educational level, both in the scholastic and in the domestic domain, opening crucial questions about the efficiency of different educational attitudes towards children and pre-adolescents.

Ethics. The study was approved by the local ethics committee (Azienda Sanitaria Locale Genovese N. 3. Ethical protocol: IIT_HRI) and the parents of all participants gave written informed consent prior to testing.

Data accessibility. The datasets and the codes supporting all the analyses and findings included in the current study, as well as the sources codes used for running the experiment, are publicly available in a dedicated OSF repository at: https://osf.io/a8s6c/. The datasets supporting the analysis included in this study contain processed, rather than raw, data, following the standards in the field.

The data are provided in the electronic supplementary material [107].

Authors' contributions. J.Z., A.S. and A.F. designed the experimental protocol. J.Z and A.F. programmed the experimental tasks. J.Z. and A.F. provided detailed instruction to the experimenter. J.Z. carried out the data analysis and wrote the manuscript. A.S. and A.F. provided suggestions for improving the manuscript.

Competing interests. The authors declare no competing interests.

Funding. The study was funded by internal funds of the Italian Institute of Technology.

Acknowledgements. We would like to thank Sonia Grasso for her valuable help with data collection.

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
