## [Peer Review File · Royal Society Open Science]

Review History

Decision letter (RSOS-201926.R0)

Dear Dr Zonca:

Manuscript ID RSOS-201926 entitled "I'm not a little kid anymore! Reciprocal social influence in child-adult interaction" which you submitted to Royal Society Open Science, has been evaluated.

I must decline the manuscript for publication in Royal Society Open Science at this time.

However, a new manuscript may be submitted which takes into consideration the comments below.

Please note that resubmitting your manuscript does not guarantee eventual acceptance, and that your resubmission will be subject to review by reviewer(s) before a decision is rendered.

You will be unable to make your revisions on the originally submitted version of your manuscript. Instead, revise your manuscript using a word processing program and save it on your computer.

You may also click the below link to start the resubmission process (or continue the process if you have already started your resubmission) for your manuscript. If you use the below link you will not be required to login to ScholarOne Manuscripts.

PLEASE NOTE: This is a two-step process. After clicking on the link, you will be directed to a webpage to confirm.

https://mc.manuscriptcentral.com/rsos?URL_MASK=3bd0fc6554034c348a5ff69c107388d7

Because we are trying to facilitate timely publication of manuscripts submitted to Royal Society Open Science, your resubmitted manuscript should be submitted by 11-May-2021. If you are unable to submit by this date please contact the Editorial Office for options.

I look forward to a resubmission.

Sincerely,
Andrew Dunn
Royal Society Open Science Editorial Office
openscience@royalsociety.org

Editor comments:

The submission was viewed by the subject editor and an associate editor. Both agreed that the task and question were interesting, but were concerned about the small number of children in each age band and lack of formal justification for power. They suggest that some power calculation is provided and that you will consider using age as a continuous variable in the analyses. This will of course not prevent you from visualising the data in age bands for presentation purposes.

Author's Response to Decision Letter for (RSOS-201926.R0)

See Appendix A.

RSOS-202124.R0

Review form: Reviewer 1

Is the manuscript scientifically sound in its present form?

Yes

Are the interpretations and conclusions justified by the results?

Yes

Is the language acceptable?

Yes

Do you have any ethical concerns with this paper?

No

Have you any concerns about statistical analyses in this paper?

No

Recommendation?

Accept with minor revision (please list in comments)

Comments to the Author(s)

This is a very nice paper that focuses on reciprocal social influence in child-adult interactions. Using an developmentally appropriate social learning task in a sample of 6, 8 and 10 year olds, the authors investigate developmental differences in the use of social information stemming from adults, as well as how this is modulated by reciprocity.

The introduction and methods are very thorough and the analyses take into account important confounding factors. I have only a few very small questions.

The 90-10% of agreement in in the experimental conditions seems somewhat extreme, it made me wonder if the authors had some data that could speak to the whether or not the participants believed the adult choices were real/honest.

Related to this I thought it may be interesting in the discussion to relate these finding to the work of Harris and colleagues on epistemic trust, and its development. Related it is interesting to see the youngest kids believe themselves to be more accurate than adults. This made me think about the literature on meta-cognition and social learning (e.g. Flemming et al papers). This might be another nice addition to an already very nice paper.

Just another speculation - in another paper by van den Bos et al., on "learning who to trust" it is shown that younger children respond particularly strong to violations of trust. Maybe it is the case that younger children no longer reciprocate after a block where the adults choose for themselves (not getting over this).

Review form: Reviewer 2

Is the manuscript scientifically sound in its present form?

Yes

Are the interpretations and conclusions justified by the results?

Yes

Is the language acceptable?

Yes

Do you have any ethical concerns with this paper?

No

Have you any concerns about statistical analyses in this paper?

No

Recommendation?

Accept with minor revision (please list in comments)

Comments to the Author(s)

This is a well-written paper which presents a novel study on reciprocal social influence in childhood. The authors used an elaborately programmed computer task to investigate the effect of susceptibility. I think the study is interesting, well-conducted, adds a valuable contribution to the current body of research and is thus worth publishing. However, I have a couple of suggestions and concerns regarding the current version of the manuscript I would like to share.

If I may comment on the first revision based on the Editors' comments:

Being trained as a Developmental Psychologist, I am aware with the typically low sample sizes in the field. What I cannot agree with is the fact that there was a one-tailed test used for the second power analysis. If the tested effect was a decades-long established effect that was shown in meta-analyses with children, I could agree going one-tailed. But the authors ground their power analysis on one study, which has been conducted with adults instead of children. The chance that this effect is either not present, or that there is a reversed effect apparent in a sample of children, is still very high! Using a one-tailed testing here is not state of the art I believe. Being an author myself, I can totally understand that authors try to do anything to prevent themselves from having to collect more data! Don't get me wrong, I personally think that the current sample size is fine regarding the typical sample sizes in the field. I just criticize the way the power analysis is conducted, as this sends a questionable message to the scientific community out there.

Regarding the age group design, I see why the authors try to keep it: We Developmentalists love these age group comparisons, as they reveal results that are straightforward and easy to communicate. Children cannot do X at age Y, but they can at age Z. Communicating a continuous increase is not as straightforward. Still, age group designs seem to have come a little out of fashion in the last years and more and more colleagues argue that using age continuously is the more correct way as it reveals more information. I believe that the authors of the current study found a nice way of combining these two ways of analysis!

Introduction:

Talking about the social influence of "peers" throughout the paper is misleading, as the actual study is not on peers but on the social influence within an adult-child-dyad. Maybe the authors could change the wording here to be more precise. Because, as they admit in the discussion: "Child interactions with peers may be indeed regulated by different mechanisms" (...as opposed to interactions with adults).

And, although I know it is written in the journal's guidelines that the judgment of the significance of the study should be left to the reader: Can the authors say a little more about why studying reciprocity in such a child-adult dyad is of interest? Of course, a study with only one participant child and an adult experimenter is more economic than a study including peer interactions. However, it would be helpful for readers to explain your theoretical rationale for choosing such an hierarchical dyad as a child-adult-dyad specifically. In addition, is there a typical, naturally occurring situation where an adult would repeatedly take over the judgment of a child and vice versa (e.g., in the parent-child or teacher-student- interactions you mention)? Such an example might help illustrating the importance of your research question and justify your chosen study design above and beyond mere economic reasons. Again, since this is not requested by the journal, please see this comment as a suggestion only.

I do not agree to the statement on the bottom of p. 3: "Fourth, including younger children (i.e., preschoolers) in a non-trivial experimental task, as the one described in this study, would have posed several issues in the comparability of results due to differences in terms of attentional focus

and performance.” Many studies in the field, including those I mention below, prove that there are age-appropriate tasks on social influences available even for preschoolers. The authors justify the chosen age groups well enough, this methodological consideration is not needed, I believe.

I was wondering why at the end of the introduction (where I would expect hypotheses) there is a summary of the results and their interpretation already? This is unusual. Maybe this is part of the journal’s guidelines and I have overseen it. Nevertheless, formulating detailed research questions and hypotheses somewhere in the paper would be helpful.

The authors present a well-selected overview of the developmental literature in the introduction. I recommend adding a couple of further papers I believe are of importance:

Add to p.2, line 31-34 and p. 3, l. 43-46:

Haun, D. B., & Tomasello, M. (2011). Conformity to peer pressure in preschool children. *Child development*, 82(6), 1759-1767.

Haun, D. B., Rekers, Y., & Tomasello, M. (2014). Children conform to the behavior of peers; other great apes stick with what they know. *Psychological science*, 25(12), 2160-2167.

On the top of page 3:

Engelmann, J. M., Over, H., Herrmann, E., & Tomasello, M. (2013). Young children care more about their reputation with ingroup members and potential reciprocators. *Developmental Science*, 16(6), 952-958.

Methods:

Do I understand correctly that there was a within-subject-design used, that is, one participant went both through the Susceptible and the Unsusceptible condition with the same adult partner? I wonder why that is. I personally would consider susceptibility to be an individual trait. As the authors say in the discussion: susceptibility as an indicator for incompetence. I wouldn’t consider that to be something that changes intrapersonally so quickly. A sudden change of the adult partner’s susceptibility during the test is unexpected and must seem strange for children (or might be understood by children, other than intended, as some “playful” change of strategy). I am sure the authors have an explanation for that they are willing to share.

I had trouble understanding which statistical analyses were conducted for which research question: At the beginning, the authors say they were using mixed-effect logistic models, whereas at the end, they said they used nonparametric Tests such as Wilcoxon throughout the paper. Maybe a specification such as: to analyze research question a, we used method b, to analyze question x, we used method y would be helpful here.

Results:

The first graph should be presented earlier for a better illustration of the written results.

In my personal opinion, the last paragraph on p. 12 would fit better in the discussion than in the results section, as it comprises a certain level of interpretation already (e.g., “took much more into consideration”, “relied significantly less on the adult’s advice”). As opposed to presenting the mere numeric differences as usual in the results section.

Just wondering: Wouldn’t one label subsamples such as these of the different age groups as small “n” instead of capital “N”? (e.g., in Figure 3)

Can you explain why condition wasn’t used as an independent variable in a mixed model design? I would have expected that. Why do you incorporate condition into a difference score

instead, using it as a dependent variable? I wonder why the authors wouldn't choose the former, more straightforward way?

Discussion:

On p. 17, the authors sum up a well-selected body of research on the cognitive abilities needed for decisions as demanded in the study. Could you please add the ages here? Because, what is missing in the discussion is an explanation why 10-year-olds might show qualitatively different behavior than 6- and 8-years-olds. Do the authors have a suggestion for an underlying mechanism that is at work in the one, but not the other age groups?

Some minor comments/ typos:

p. 2, line 17: "3/4 years of age" reads like 0.75, maybe change to "3 to 4 years of age"

p. 7, l. 14: there is an "s" missing after participant

p. 9, l. 47: the "i" of influence should be in italics.

First line of the discussion: "behavioral" should be without the -al, or there is a noun missing

p. 16, l. 26: "children did reciprocated", omit the "d"

I am not a native speaker, but based on the literature, "10 years old children" seems uncommon, wouldn't one say "10-year-old children" instead, without the "s"?

Decision letter (RSOS-202124.R0)

Dear Dr Zonca

On behalf of the Editors, we are pleased to inform you that your Manuscript RSOS-202124 "I'm not a little kid anymore! Reciprocal social influence in child-adult interaction" has been accepted for publication in Royal Society Open Science subject to minor revision in accordance with the referees' reports. Please find the referees' comments along with any feedback from the Editors below my signature.

Please submit your revised manuscript and required files (see below) no later than 7 days from today's (ie 06-Jul-2021) date. Note: the ScholarOne system will 'lock' if submission of the revision is attempted 7 or more days after the deadline. If you do not think you will be able to meet this deadline please contact the editorial office immediately.

on behalf of Dr Teodora Gliga (Associate Editor) and Essi Viding (Subject Editor)
openscience@royalsociety.org

Associate Editor Comments to Author (Dr Teodora Gliga):

I have now received comments from two experts in the field. They both find your study well conducted and the questions addressed important. One of the reviewers requests further methodological clarifications as well as listing your hypothesis at the end of the introduction. Both reviewers recommend further literature to integrate into your introduction/discussion. I therefore invite you to respond to these comments and revise your paper accordingly.

Reviewer comments to Author:

Reviewer: 1

Comments to the Author(s)

This is a very nice paper that focuses on reciprocal social influence in child-adult interactions. Using an developmentally appropriate social learning task in a sample of 6, 8 and 10 year olds, the authors investigate developmental differences in the use of social information stemming from adults, as well as how this is modulated by reciprocity.

The introduction and methods are very thorough and the analyses take into account important confounding factors. I have only a few very small questions.

The 90-10% of agreement in in the experimental conditions seems somewhat extreme, it made me wonder if the authors had some data that could speak to the whether or not the participants believed the adult choices were real/honest.

Related to this I thought it may be interesting in the discussion to relate these finding to the work of Harris and colleagues on epistemic trust, and its development. Related it is interesting to see the youngest kids believe themselves to be more accurate than adults. This made me think about the literature on meta-cognition and social learning (e.g. Flemming et al papers). This might be another nice addition to an already very nice paper.

Just another speculation - in another paper by van den Bos et al., on "learning who to trust" it is shown that younger children respond particularly strong to violations of trust. Maybe it is the case that younger children no longer reciprocate after a block where the adults choose for themselves (not getting over this).

Reviewer: 2

Comments to the Author(s)

This is a well-written paper which presents a novel study on reciprocal social influence in childhood. The authors used an elaborately programmed computer task to investigate the effect of susceptibility. I think the study is interesting, well-conducted, adds a valuable contribution to the current body of research and is thus worth publishing. However, I have a couple of suggestions and concerns regarding the current version of the manuscript I would like to share.

If I may comment on the first revision based on the Editors' comments:

Being trained as a Developmental Psychologist, I am aware with the typically low sample sizes in the field. What I cannot agree with is the fact that there was a one-tailed test used for the second power analysis. If the tested effect was a decades-long established effect that was shown in meta-analyses with children, I could agree going one-tailed. But the authors ground their power analysis on one study, which has been conducted with adults instead of children. The chance that this effect is either not present, or that there is a reversed effect apparent in a sample of children, is still very high! Using a one-tailed testing here is not state of the art I believe. Being an author myself, I can totally understand that authors try to do anything to prevent themselves from having to collect more data! Don't get me wrong, I personally think that the current sample size is fine regarding the typical sample sizes in the field. I just criticize the way the power analysis is conducted, as this sends a questionable message to the scientific community out there.

Regarding the age group design, I see why the authors try to keep it: We Developmentalists love these age group comparisons, as they reveal results that are straightforward and easy to communicate. Children cannot do X at age Y, but they can at age Z. Communicating a continuous increase is not as straightforward. Still, age group designs seem to have come a little out of fashion in the last years and more and more colleagues argue that using age continuously is the more correct way as it reveals more information. I believe that the authors of the current study found a nice way of combining these two ways of analysis!

Introduction:

Talking about the social influence of "peers" throughout the paper is misleading, as the actual study is not on peers but on the social influence within an adult-child-dyad. Maybe the authors could change the wording here to be more precise. Because, as they admit in the discussion: "Child interactions with peers may be indeed regulated by different mechanisms" (...as opposed to interactions with adults).

And, although I know it is written in the journal's guidelines that the judgment of the significance of the study should be left to the reader: Can the authors say a little more about why studying reciprocity in such a child-adult dyad is of interest? Of course, a study with only one participant child and an adult experimenter is more economic than a study including peer interactions. However, it would be helpful for readers to explain your theoretical rationale for choosing such an hierarchical dyad as a child-adult-dyad specifically. In addition, is there a typical, naturally occurring situation where an adult would repeatedly take over the judgment of a child and vice versa (e.g., in the parent-child or teacher-student- interactions you mention)? Such an example might help illustrating the importance of your research question and justify your chosen study design above and beyond mere economic reasons. Again, since this is not requested by the journal, please see this comment as a suggestion only.

I do not agree to the statement on the bottom of p. 3: "Fourth, including younger children (i.e., preschoolers) in a non-trivial experimental task, as the one described in this study, would have posed several issues in the comparability of results due to differences in terms of attentional focus and performance." Many studies in the field, including those I mention below, prove that there are age-appropriate tasks on social influences available even for preschoolers. The authors justify the chosen age groups well enough, this methodological consideration is not needed, I believe.

I was wondering why at the end of the introduction (where I would expect hypotheses) there is a summary of the results and their interpretation already? This is unusual. Maybe this is part of the journal's guidelines and I have overseen it. Nevertheless, formulating detailed research questions and hypotheses somewhere in the paper would be helpful.

The authors present a well-selected overview of the developmental literature in the introduction. I recommend adding a couple of further papers I believe are of importance:

Add to p.2, line 31-34 and p. 3, l. 43-46:

Haun, D. B., & Tomasello, M. (2011). Conformity to peer pressure in preschool children. *Child development*, 82(6), 1759-1767.

Haun, D. B., Rekers, Y., & Tomasello, M. (2014). Children conform to the behavior of peers; other great apes stick with what they know. *Psychological science*, 25(12), 2160-2167.

On the top of page 3:

Engelmann, J. M., Over, H., Herrmann, E., & Tomasello, M. (2013). Young children care more about their reputation with ingroup members and potential reciprocators. *Developmental Science*, 16(6), 952-958.

Methods:

Do I understand correctly that there was a within-subject-design used, that is, one participant went both through the Susceptible and the Unsusceptible condition with the same adult partner? I wonder why that is. I personally would consider susceptibility to be an individual trait. As the authors say in the discussion: susceptibility as an indicator for incompetence. I wouldn't consider that to be something that changes intrapersonally so quickly. A sudden change of the adult partner's susceptibility during the test is unexpected and must seem strange for children (or might be understood by children, other than intended, as some "playful" change of strategy). I am sure the authors have an explanation for that they are willing to share.

I had trouble understanding which statistical analyses were conducted for which research question: At the beginning, the authors say they were using mixed-effect logistic models, whereas at the end, they said they used nonparametric Tests such as Wilcoxon throughout the paper. Maybe a specification such as: to analyze research question a, we used method b, to analyze question x, we used method y would be helpful here.

Results:

The first graph should be presented earlier for a better illustration of the written results.

In my personal opinion, the last paragraph on p. 12 would fit better in the discussion than in the results section, as it comprises a certain level of interpretation already (e.g., "took much more into consideration", "relied significantly less on the adult's advice"). As opposed to presenting the mere numeric differences as usual in the results section.

Just wondering: Wouldn't one label subsamples such as these of the different age groups as small "n" instead of capital "N"? (e.g., in Figure 3)

Can you explain why condition wasn't used as an independent variable in a mixed model design? I would have expected that. Why do you incorporate condition into a difference score instead, using it as a dependent variable? I wonder why the authors wouldn't choose the former, more straightforward way?

Discussion:

On p. 17, the authors sum up a well-selected body of research on the cognitive abilities needed for decisions as demanded in the study. Could you please add the ages here? Because, what is missing in the discussion is an explanation why 10-year-olds might show qualitatively different behavior than 6- and 8-years-olds. Do the authors have a suggestion for an underlying mechanism that is at work in the one, but not the other age groups?

Some minor comments/ typos:

p. 2, line 17: "3/4 years of age" reads like 0.75, maybe change to "3 to 4 years of age"

p. 7, l. 14: there is an "s" missing after participant

p. 9, l. 47: the "i" of influence should be in italics.

First line of the discussion: "behavioral" should be without the -al, or there is a noun missing

p. 16, l. 26: "children did reciprocated", omit the "d"

I am not a native speaker, but based on the literature, "10 years old children" seems uncommon, wouldn't one say "10-year-old children" instead, without the "s"?

===PREPARING YOUR MANUSCRIPT===

===PREPARING YOUR REVISION IN SCHOLARONE===

Author's Response to Decision Letter for (RSOS-202124.R0)

See Appendix B.

Decision letter (RSOS-202124.R1)

Dear Dr Zonca,

I am pleased to inform you that your manuscript entitled "I'm not a little kid anymore! Reciprocal social influence in child-adult interaction" is now accepted for publication in Royal Society Open Science.

on behalf of Dr Teodora Gliga (Associate Editor) and Essi Viding (Subject Editor)
openscience@royalsociety.org

Appendix A

Response to decision letter, manuscript ID RSOS-201926

We would like to thank the subject editor and the associate editor for their insightful and valuable comments. We have implemented their recommendations and wish to submit a revised version of the manuscript for further consideration in Royal Society Open Science.

Editor comments:

The submission was viewed by the subject editor and an associate editor. Both agreed that the task and question were interesting, but were concerned about the small number of children in each age band and lack of formal justification for power. They suggest that some power calculation is provided and that you will consider using age as a continuous variable in the analyses. This will of course not prevent you from visualising the data in age bands for presentation purposes.

Authors' response:

We understand the importance of including in the manuscript a formal justification for power concerning our sample size in the experiment. We also agree that using age as a continuous variable could be an efficient way to increase the robustness and generalizability of our analyses given a limited sample size. We had already taken into consideration this approach during the preparation of the original manuscript and, at that time, we had decided to include only group-based analyses since we did not want to make the result overly complex and redundant, and given that these group-based analyses reveal crucial features underlying the emergence of reciprocal social influence in child-adult interaction. In this respect, we acknowledge that analyzing our data using age either as a continuous or a categorical variable entails two different facets of our main experimental hypothesis: the former aims at investigating the existence of an increase in reciprocal behavior along with age; the latter allows to test the specific hypothesis that reciprocal behavior in social influence emerges at a specific step of child development. Since, to our knowledge, no studies to date have investigated the developmental features of reciprocity in social influence contexts, in the revised version of the manuscript we use both approaches to better characterize the emergence of reciprocal social influence during childhood and, at the same time, add robustness and generalizability to our findings. We have also provided a formal justification of the selected sample size for both types of statistical analyses, which can be found in the main manuscript in the Methods/Participants section (page 4). This is the part of the section concerning the power analysis:

“...The target sample size was estimated based on the predicted statistical analyses aimed to detect 1) a significant effect of age on reciprocity, defined as the differences between the influence exerted by the adult partner on children in *Susceptible* and *Unsusceptible* conditions and 2) a significant within-subject effect of condition (*Susceptible* or *Unsusceptible*) on the influence exerted by adult's partner influence on children in each of the three age cohorts (6, 8, 10 years old). The two analyses are aimed at exploring two different facets of our main experimental hypothesis: the former aims at testing the existence of a linear increase in the occurrence of reciprocal behavior along with age, whereas the latter aims at investigating the emergence of reciprocal behavior at a specific step of child development.

Concerning the former analysis, we planned to run a multiple regression analysis with reciprocity as dependent variable and age as continuous independent variable, adjusting for potential confound variables such as performance in perceptual estimation, individual mean social influence and performance ratings. Importantly, to our knowledge there are no previous results showing an effect of age on reciprocal social influence and therefore we do not have any priors about the magnitude of the hypothesized effects. We have therefore assumed a classical medium effect size ($f^2 = 0.15$) along with a two-tailed distribution and a

power of 0.8 to obtain the required sample size for a multiple linear regression analysis. The required total sample size resulting from this analysis is 55.

Concerning the second type of analysis, we have referred to a recent study [69] investigating the emergence of reciprocal social influence in adults. In two experiments, the authors have shown that the adults' willingness to follow the opinion of an (alleged) adult partner was higher when the partner herself had shown high consideration (Susceptible condition) rather than low consideration (Unsusceptible condition) towards them. The effect size of the observed within-subject effect of condition (Susceptible – Unsusceptible) was considerably large in both experiments (Exp. 1: $r = 0.57$, $d = 1.39$; Exp. 2: $r = 0.46$, $d = 1.03$. See the "Statistical data analysis" paragraph for computation of effect sizes). For this reason, we have assumed a medium-to-large effect size ($d = 0.6$) to test the emergence of an effect of reciprocal social influence (i.e., effect of condition) at specific steps of development. Based on these solid previous results, we have also assumed a one-tailed distribution to test a specific directional hypothesis about the emergence of reciprocity in children (i.e., influence in the *Susceptible* condition higher than influence in the *Unsusceptible* condition). Given these assumptions, along with an assumed power of 0.8, the recommended sample size for a Wilcoxon signed-rank test is 20 (See the "Statistical data analysis" paragraph for selection of the statistical methods and tests supporting hypothesis testing). We therefore planned to collect data of at least 20 children for each age cohort, compatibly with the demands of the selected elementary school. This requirement is in line with the requirements of the previous analysis (i.e., a total of 55 participants across age bands)."

We have also expanded the "Statistical data analysis" section (page 9) including additional information about the utilization of age as a continuous variable in regression analyses. In particular, we have specified that:

"... throughout the paper we also used individual variables to better characterize between-subject behavioral heterogeneity. In this respect, the two most important dependent variables we analysed were: 1) *influence*, defined as the proportion of trials in which participants chose their partner's estimate in their final decisions and 2) *reciprocity*, computed as the difference between influence in Susceptible and Unsusceptible conditions for each participant. These dependent variables have been either compared across experimental conditions and groups or analyzed in relation with several predictors including age, participants' perceptual error, agents' response distance and performance ratings. Participant's age (when treated as a continuous variable) was computed as ' months of age / 12..."

We have therefore reported some new findings, in the results section, showing an effect of age (used as a continuous variable) on 1) the influence exerted by the adult partner on children and 2) reciprocal social influence. Here we report the manuscript section reporting these analyses:

- 1) (Page 10). "... We show a significant relationship between participants' influence, computed as the proportion of trials in which the child chose the adult's response in her final decisions, and age (linear regression with robust standard errors, standardized coefficient (β) = 0.52, unstandardized coefficient (B) = 0.06, $t = 4.79$, $p < 0.001$). This result does not change when adjusting for agents' response distance (effect of age: $\beta = 0.52$, B = 0.06, $t = 3.80$, $p < 0.001$; effect of distance: $\beta = -0.01$, B = - 0.01, $t = - 0.04$, $p = 0.967$) or participants' estimation error (effect of age: $\beta = 0.52$, B = 0.06, $t = 3.64$, $p = 0.001$; effect of error: $\beta = - 0.01$, B = - 0.01, $t = - 0.05$, $p = 0.964$)."
- 2) (Page 14). "... First, we found that reciprocity, defined as the difference between participants' influence in the Susceptible and the Unsusceptible condition, is significantly predicted by age (multiple linear regression with robust standard errors, $\beta = 0.43$, B = 0.04, $t = 2.25$, $p = 0.029$), while it is not explained by potential confound variables such as participants' estimation error ($\beta =$

0.16, $B = 0.12$, $t = 1.44$, $p = 0.157$), average influence ($\beta = -0.27$, $B = -0.24$, $t = -1.73$, $p = 0.089$) and performance ratings ($\beta = 0.13$, $B = 0.01$, $t = 0.81$, $p = 0.424$).”

Nevertheless, we acknowledge the importance of reporting results that use *age* as a group factor, along with the above-mentioned regression analyses, to appreciate a better characterization of the developmental trajectory of reciprocal social influence. More specifically, in all the main analyses of the manuscript we observe a marked difference between the behavior of 10 years old children and the two younger participants' cohorts, both from an informational (i.e., analyses on general social influence) and normative (i.e., analysis on reciprocity) point of view. The observed qualitative and quantitative differences underlying the behavior of 10 years old children would be impossible to disclose just using age as a continuous variable.

We hope that the inclusion of a formal justification for power will address the editors' concerns about the group-based analyses and that, at the same time, the introduction of analyses using age as a continuous variable will add robustness and generalizability to our findings.

Thank you for considering our re-submission.

Sincerely,

Joshua Zonca, Anna Folsø and Alessandra Sciutti

Appendix B

Response to Editors and reviewers

Manuscript RSOS-202124

I'm not a little kid anymore! Reciprocal social influence in child-adult interaction

Joshua Zonca, Anna Folso, Alessandra Sciutti

We would like to thank the Editors and the reviewers for their insightful and valuable comments, which helped us to improve significantly our manuscript. In the revised manuscript, we rewrote and restructured extensive parts of Introduction, Methods and Discussion sections, following Editors' and reviewers' suggestions. In particular:

- We have integrated the recommended literature in Introduction and Discussion sections. The suggested literature was actually important for providing a more exhaustive interpretation of our results.
- We have clarified and reorganized our hypotheses at the end of the Introduction and in a dedicated paragraph of the Methods section ("Research questions and statistical data analysis"), where we have also restructured the description of our statistical analyses.
- We have followed the suggestions of reviewer#2 concerning the sample size estimation and provided a more detailed and comprehensive description of these power analyses in a specific paragraph at the end of the Methods section ("Sample size estimation").
- We have clarified the reasons underlying some of the choices of our experimental design in Introduction and Discussion sections.

All the changes were highlighted in yellow in the main manuscript ("Manuscript with tracked changes" version). In the current document, we refer the above-mentioned version of the manuscript ("Manuscript with tracked changes") when referring to specific changes in the manuscript and relative pages.

Below we report the detailed replies to all reviewers' comments. The text of the comments is in italic and the replies are in normal font.

Reviewer Comments:

Reviewer 1

Summary and general comments:

This is a very nice paper that focuses on reciprocal social influence in child-adult interactions. Using an developmentally appropriate social learning task in a sample of 6, 8 and 10 year olds, the authors investigate developmental differences in the use of social information stemming from adults, as well as how this is modulated by reciprocity. The introduction and methods are very thorough and the analyses take into account important confounding factors. I have only a few very small questions.

Response

We thank the reviewer for their positive assessment of our work and for their insightful and useful comments.

Comment#1.1

The 90-10% of agreement in in the experimental conditions seems somewhat extreme, it made me wonder if the authors had some data that could speak to the whether or not the participants believed the adult choices were real/honest.

Response#1.1

We thank the reviewer for the thoughtful comment. The choice of implementing such extreme behaviors in the two conditions is motivated by the need to make the adult's behavior clearly distinguishable across conditions in the eyes of children. Indeed, we are convinced that a less extreme between-condition difference in terms of adult partner's probability distributions (e.g., 70% and 30% of agreement) would have been much more difficult to grasp, understand and mentally represent by children, especially by the youngest. This would have casted doubt on the comparability of our data and our experimental manipulation across age groups due to potential differences in the basic understanding of the statistical and distributional choice patterns characterizing the adult partner's behavior.

To prevent the potential effect hypothesized by the author, we actually chose to implement the "inconsistent" choice of the adult partner (final decision on own response in the Susceptible condition, final decision on the child's response in the Unsusceptible condition) during the second observation turn in each condition (See Figure S2 in the Supplementary Information). We reasoned that placing the inconsistent choice at this early stage of the task would have prevented children from assuming, at the very beginning of the task, that the adult's choice were in some way "fixed" and the partner was not actually choosing based on individual and contingent considerations. We implemented the inconsistent choice also at the second observation turn in the second condition (following the temporal order of presentation), again to avoid to give the impression of an abrupt change in the adult's behavior.

We highlight that the current task is a simplified version of a task used in adults dyads and triads (Mahmoodi et al., 2018 [73]: Experiment 2a; Zonca et al., 2021[74]: Experiment 1). Also in these works, participants interacted with an alleged agent that changed behavior from susceptible to unsusceptible (or viceversa) at a certain stage of the experiment, following a within-subject block design similar to the one implemented in the current work (with a brief transition block between the two conditions, to make less evident the abrupt behavioral shift). The only main difference was that participants' and partner's final decisions, in the two above-mentioned studies, were not dichotomous: participants' and their partners could select any position between own and other's perceptual estimate and the shift from their original estimate towards that of the partner was taken as an index of social influence. Although this detail somehow increases the perceived variability across trials in the partner's final decisions, we acknowledge that the partner's average social influence in "observation" turns in these studies was markedly difference across conditions and rather extreme for the Unsusceptible condition (Mahmoodi et al: 7% in the Unsusceptible condition and 50% in the Susceptible one; Zonca et al., 2021: 18% in the Unsusceptible condition and 64% in the Susceptible one). Talking about the study of our group (Zonca et al., 2021), during the final debriefing all participants confirmed that did not realize that choices were not implemented by a real human partner and they did not spot any strange behavior in the alleged partner. These recent results on adult-adult interaction had convinced us that young children would be unlikely to generate the idea that the choices implemented by the adult partner were pre-determined, especially if we consider they were completely naïve in respect to experimental research and relative expedients.

In this regard, we highlight that none of the children explicitly expressed doubts about the choices of the adult partner, neither before nor during nor after the experiment.

Comment#1.2

Related to this I thought it may be interesting in the discussion to relate these finding to the work of Harris and colleagues on epistemic trust, and its development. Related it is interesting to see the youngest kids believe themselves to be more accurate than adults. This made me think about the literature on meta-cognition and social learning (e.g. Flemming et al papers). This might be another nice addition to an already very nice paper.

Response#1.2

We thank reviewer#1 for the relevant and insightful suggestions. We believe that these suggestions improved significantly our manuscript. Actually, we believe that the two streams of research suggested by the reviewer are particularly relevant to our work: therefore, we elaborated on these themes in the revised

version of the Manuscript, referring to literature on epistemic trust and meta-cognition in the Discussion. Discussion on the work by Harris and colleagues on epistemic trust helped us in enriching the background on early developmental mechanisms of selective social learning in infants, toddlers and children. Moreover, the literature on metacognition offered a solid background for the interpretation of results on social influence in children of different ages (e.g, the reason why 6-year-old children trust the adult partner less than 10-year-olds). These additions have been included in the Discussion section of the revised manuscript (see the relevant, highlighted section of text in the Discussion at pages 17-18 and references 75, 84-91 in the References section of the revised manuscript (“Manuscript with tracked changes” version)).

Comment#1.3

Just another speculation - in another paper by van den Bos et al., on "learning who to trust" it is shown that younger children respond particularly strong to violations of trust. Maybe it is the case that younger children no longer reciprocate after a block where the adults choose for themselves (not getting over this).

Response#1.3. We thank the reviewer for the interesting suggestion. Actually, this is an interesting and relevant paper that we have included in the revised version of the manuscript (Introduction, page 1, reference 55). The hypothesis advanced by the reviewer is intriguing. We have run a specific analysis to test it. We considered the order of presentation of the two conditions (Susceptible first or Unsusceptible first) to characterize potential reciprocity effects driven by the previous condition. We ran a regression with reciprocity (mean influence in the Susceptible condition – mean influence in the Unsusceptible condition) as dependent variable, and age group, order of conditions (Susceptible first or Unsusceptible first) and their interactions as independent factors. If the hypothesis about the violation of trust is correct, we should observe a higher effect of (positive) reciprocity (for a specific age cohort) when children start with the Susceptible condition. On the contrary, if they start the experiment with the Unsusceptible condition, they should reciprocate relatively less in the Susceptible condition. The analysis does not return any significant result. Also by simply looking at the regression betas and descriptive statistics, we do not see any hint about a potential effect in the direction proposed by the reviewer. Nevertheless, we acknowledge that this analysis is rather underpowered, given that the order of conditions is a between-subject factor, which halves the number of participants per condition in the regression. For this reason, we did not include this analysis in the revised version of the manuscript. We hope that future studies may deepen the temporal and endogenous dynamics of reciprocal social influence to understand the impact of previous violations of trust or social norms in reciprocity of social influence.

Reviewer 2

Summary and general comments:

This is a well-written paper which presents a novel study on reciprocal social influence in childhood. The authors used an elaborately programmed computer task to investigate the effect of susceptibility. I think the study is interesting, well-conducted, adds a valuable contribution to the current body of research and is thus worth publishing. However, I have a couple of suggestions and concerns regarding the current version of the manuscript I would like to share.

Response

We thank reviewer#2 for the positive assessment of our work and for their insightful and thoughtful comments.

Comment#2.1

If I may comment on the first revision based on the Editors' comments: Being trained as a Developmental Psychologist, I am aware with the typically low sample sizes in the field. What I cannot agree with is the

fact that there was a one-tailed test used for the second power analysis. If the tested effect was a decades-long established effect that was shown in meta-analyses with children, I could agree going one-tailed. But the authors ground their power analysis on one study, which has been conducted with adults instead of children. The chance that this effect is either not present, or that there is a reversed effect apparent in a sample of children, is still very high! Using a one-tailed testing here is not state of the art I believe. Being an author myself, I can totally understand that authors try to do anything to prevent themselves from having to collect more data! Don't get me wrong, I personally think that the current sample size is fine regarding the typical sample sizes in the field. I just criticize the way the power analysis is conducted, as this sends a questionable message to the scientific community out there.

Response#2.1

We thank the reviewer for the thoughtful and important comment. Actually, we understand the reviewer's position on the one-tailed test in the second sample size estimation analysis. We agree that one-tailed effects should not be assumed without a strong a priori assumption about the direction of the hypothesized effect. We also agree that reference to a single previous study on the topic of investigation does not generally fulfill these requirements. In fact, our choice of selecting this approach for the first version of the manuscript stems from a single article (Mahmoodi et al., 2018 [73]) investigating the same topic in adult-adult dyads. We highlight that the same article also presented a control experiment (Exp. 2a) in which participants interact with a computer. Moreover, the same design has been used in a recently published work of our group (Zonca et al., 2021, now in the References [74]) investigating reciprocal social influence in adult triads, with a similar control condition focusing on human-computer interaction. In both these papers, we observe an effect of reciprocity in the human condition, but we do not observe an *anti-reciprocity* effect in the computer condition. In other words, when interacting with a computer, the effect of reciprocity disappears, but does not reverse. These results had originally convinced us that could be safe enough to assume that an anti-reciprocity effect would be very unlikely, even in children, since it does not even arise when interacting with a computer.

Nonetheless, we understand the position of the reviewer: this assumption is probably not sufficiently corroborated by previous literature, and the statistical approach used in the above-mentioned analysis is not state of the art. Also, as correctly suggested by the reviewer, we are testing reciprocal social influence in a completely new population, in which social norms might be underdeveloped, possibly leading to a reversal of the effect. Given these considerations, in the revised version of the manuscript we followed the reviewer's suggestion and assumed a two-tailed test, as in the first power analysis. For the same reason, the relative statistical analysis in the manuscript used a two-tailed Wilcoxon signed-rank test, as all the other analyses in the paper. Moreover, we generally agree that the power analysis (and the relative assumptions and motivations) should have been reported in a more exhaustive, precise and detailed fashion. In this regard, we actually re-structured the Methods section to account for these changes and those suggested in comment #2.8 by the same reviewer. In the revised manuscript, we provide a more detailed and structured "Research questions and statistical data analysis" section (pages 8-10, "Manuscript with tracked changes"), which precedes a new dedicated section on sample size estimation named "Sample size estimation" (pages 10-11). At the beginning of this section, we also acknowledged that these power analyses aimed at validating a sample size that was already limited due to the demands of the elementary school in which the study was conducted. Indeed, the school gave us the availability to test students in three classes at three different educational levels. Each class was composed of 20-25 children: this range reflected an upper bound for our potential sample size. Therefore, we acknowledged and highlighted that the two different power analyses aimed at validating the robustness of this sample size for the predicted statistical analyses. Following this approach, we put more attention on the identification of the assumed effect size, which has been rather arbitrary in the previous version. Indeed, in the original version we had referred to an effect size of about $d = 1.20$ from Mahmoodi et al. (2018), and we assumed an effect size halved in magnitude for our developmental sample, consistently with a medium-to-large effect. However, as suggested by the reviewer, this single paper might be non-sufficient to characterize the estimated effect size for the power analysis. In the revised version of the manuscript, we aimed at better characterizing the assumed effect size. We reported

calculation of the minimum effect of *condition* observable in our dependent variable (*influence*). Given that children face 10 *decision* trials for condition, and *influence* is calculated as the proportion of trials in which participants chose their partner's estimate in their final decisions, the minimum within-subject difference we can expect is 0.1, consistent with a difference of one choice across conditions. Assuming an average standard deviation of 0.15 in the difference in *influence* across conditions, we can compute the expected effect size, which is 0.67. This is actually in line with the assumption of a medium ($d = 0.5$) to large ($d = 0.8$) effect size, as stated in the original manuscript. At this point, we reported the effect sizes of the main effects of Mahmoodi and colleagues, to corroborate the validity of our estimation and assumptions about the effect size. Then we repeated our original power analysis on Wilcoxon signed-rank test, assuming an effect size of 0.67, a power of 0.8 and a two-tailed test. The result of this analysis is an estimated sample of 21 participants in each group, which is in line with the requirements of the power analysis using age as continuous variable (i.e., 55 participants in total), as well as with our actual sample size.

Comment#2.2

Regarding the age group design, I see why the authors try to keep it: We Developmentalists love these age group comparisons, as they reveal results that are straightforward and easy to communicate. Children cannot do X at age Y, but they can at age Z. Communicating a continuous increase is not as straightforward. Still, age group designs seem to have come a little out of fashion in the last years and more and more colleagues argue that using age continuously is the more correct way as it reveals more information. I believe that the authors of the current study found a nice way of combining these two ways of analysis!

Response#2.2

We thank the reviewer for the thoughtful consideration. Following the original Editors' suggestions, we have tried to combine both data analysis approaches to return the highest amount of information from our data and, at the same time, send clear messages to the audience.

Comment#2.3

Introduction:

Talking about the social influence of "peers" throughout the paper is misleading, as the actual study is not on peers but on the social influence within an adult-child-dyad. Maybe the authors could change the wording here to be more precise. Because, as they admit in the discussion: "Child interactions with peers may be indeed regulated by different mechanisms" (...as opposed to interactions with adults). And, although I know it is written in the journal's guidelines that the judgment of the significance of the study should be left to the reader: Can the authors say a little more about why studying reciprocity in such a child-adult dyad is of interest? Of course, a study with only one participant child and an adult experimenter is more economic than a study including peer interactions. However, it would be helpful for readers to explain your theoretical rationale for choosing such a hierarchical dyad as a child-adult-dyad specifically. In addition, is there a typical, naturally occurring situation where an adult would repeatedly take over the judgment of a child and vice versa (e.g., in the parent-child or teacher-student- interactions you mention)? Such an example might help illustrating the importance of your research question and justify your chosen study design above and beyond mere economic reasons. Again, since this is not requested by the journal, please see this comment as a suggestion only.

Response#2.3

We thank the reviewer for the insightful considerations. We totally agree that our work targets a specific type of dyadic, hierarchical interaction between a child and an adult partner and that our results do not necessarily extend to child-child interactions, as already pointed out in the Discussion section. References to "peers" in the main manuscript generally concerned results on adult-adult or child-child interaction coming from the literature review, or associations with real-life peer interactions. Nonetheless, we understand that referring to peer interaction, in some parts of the manuscript, may be misleading in the

sense that shifts the focus on types of interactions that have not been targeted in the current work. Following the reviewer's advice, we changed the wording in some parts of the manuscript to account for this issue. We also agree that the choice of using a child-adult interaction might be justified and explained a bit more along the manuscript. Besides practical and economic advantages, we believe this kind of interaction to be crucial in several real-life scenarios, especially in educational contexts both in domestic (i.e. parent-child relationship) and scholastic (i.e., teacher-student relationship) environments. We know that children receive an impressive amount of novel information from adults during development, which sometimes conflicts with children's current knowledge. When children's beliefs and information coming from adults are in conflict, children may be unwilling to accept novel information from adult informants (see, for, instance Koenig & Harris, 2007 [75]), which interferes with the ongoing learning processes. In this context, children might require a certain degree of consideration from adults in order to accept this new information, following the reciprocal mechanisms of social influence observed in adult-adult interactions (Mahmoodi et al., 2018 [73]; Zonca et al., 2021[74]). This may be especially true at later stages of development, since we know that, during development, children become more and more sensitive to normative principles (e.g., reciprocity). Following the suggestion of reviewer#2, we elaborated on the significance of this specific relational, hierarchical social context in the Introduction section, right before introducing the experimental task (see the highlighted text at page 3).

In general, we believe that both child-adult and child-child interactive contexts are extremely interesting for the theme of development of reciprocal social influence. Indeed, we are already planning an experiment on child-child interaction on these themes: we believe that a new study on "peer" interaction in children may offer new insights on the development of reciprocity in social influence and complement the results of the current work.

Comment#2.4

I do not agree to the statement on the bottom of p. 3: "Fourth, including younger children (i.e., preschoolers) in a non-trivial experimental task, as the one described in this study, would have posed several issues in the comparability of results due to differences in terms of attentional focus and performance." Many studies in the field, including those I mention below, prove that there are age-appropriate tasks on social influences available even for preschoolers. The authors justify the chosen age groups well enough, this methodological consideration is not needed, I believe.

Response#2.4

We thank the reviewer for their suggestion; we have deleted this statement in the revised version of the manuscript.

Comment#2.5

I was wondering why at the end of the introduction (where I would expect hypotheses) there is a summary of the results and their interpretation already? This is unusual. Maybe this is part of the journal's guidelines and I have overseen it. Nevertheless, formulating detailed research questions and hypotheses somewhere in the paper would be helpful.

Response#2.5

We thank reviewer #2 for the useful suggestion. We have now changed the last part of the Introduction, deleting the summary of the results and providing a summary of the main hypotheses and research questions (page 5). A detailed description of research questions and hypotheses can be found in a new dedicated section called "Research questions and statistical data analysis", pages 8-10 (see also Response #2.8).

Comment#2.6

The authors present a well-selected overview of the developmental literature in the introduction. I recommend adding a couple of further papers I believe are of importance:

Add to p.2, line 31-34 and p. 3, l. 43-46:

*Haun, D. B., & Tomasello, M. (2011). Conformity to peer pressure in preschool children. *Child development*, 82(6), 1759-1767.*

*Haun, D. B., Rekers, Y., & Tomasello, M. (2014). Children conform to the behavior of peers; other great apes stick with what they know. *Psychological science*, 25(12), 2160-2167.*

On the top of page 3:

*Engelmann, J. M., Over, H., Herrmann, E., & Tomasello, M. (2013). Young children care more about their reputation with ingroup members and potential reciprocators. *Developmental Science*, 16(6), 952-958.*

Response#2.6.

We thank the reviewer for the useful suggestions; we have added the suggested papers in the revised version of the manuscript (references 42, 43, 63).

Comment#2.7

Methods:

Do I understand correctly that there was a within-subject-design used, that is, one participant went both through the Susceptible and the Unsusceptible condition with the same adult partner? I wonder why that is. I personally would consider susceptibility to be an individual trait. As the authors say in the discussion: susceptibility as an indicator for incompetence. I wouldn't consider that to be something that changes intrapersonally so quickly. A sudden change of the adult partner's susceptibility during the test is unexpected and must seem strange for children (or might be understood by children, other than intended, as some "playful" change of strategy). I am sure the authors have an explanation for that they are willing to share.

Response#2.7

We thank the reviewer for the thoughtful comment. As correctly understood by the reviewer, we used a within-subject design consisting in a behavioral change of the same adult partner across conditions. We highlight that this kind of design has been already used in the two (and only) studies that have recently investigated reciprocal social influence in adults (Mahmoodi et al., 2018 [73]: Experiment 2a; Zonca et al., 2021[74]: Experiment 1). The idea behind this kind of design, besides practical and economic advantages, lies in the dynamic nature of reciprocity. Reciprocity is a social norm that regulates human behavior in repeated interactions and primarily aims at supporting continuous collaboration and cooperation and preventing defective behavior (i.e., one of the interacting partner(s) egoistically maximizes their own utility at others' expenses, taking advantage of their vulnerability). In these terms, reciprocity sustains dynamic interactions and is generally studied (e.g., in behavioral economics) as a process regulating the rise and fall of cooperative (or pro-social) behavior during interaction *with the same partner*.

However, the dynamic principles underlying our experimental task are more complex than usual tasks investigating reciprocity (e.g., repeated and multi-stage games). Participants do not have to implement a simple binary choice (e.g., cooperate or defect), but have to combine informational properties of the decision context (i.e., perceived accuracies of own and other's response) with normative ones (previous own and other's final decisions) to make final decisions. In this context, trial-by-trial analyses based on the previous partner's move (as usually done, for instance, in repeated or multi-stage games in behavioral economics) are very difficult and noisy because of confounding factors (i.e., the informational characteristics of the current trial, participants' confidence in their current own response, etc.). Therefore, one of the easiest and cleanest ways to study dynamic modulation of social influence by reciprocity is

implementing within-subject design as the ones implemented in the current and previous works investigating the same topic [73, 74]. We agree with the reviewer that we refer to a specific type of child-adult interaction in terms of normative principles: children interact with someone that does not trust their opinions and then, suddenly, starts to trust them (or vice-versa). We acknowledge that this is just one of the several potential interactive settings and one of the many ways to manipulate the partner's susceptibility. For instance, it may be interesting to understand how children react to two different adult partners (a "susceptible" partner and an "unsusceptible" one), as suggested by the reviewer. It may be the case that this static behavior is easier to understand for younger children, who may show a clearer differentiation between the two partners and then adapt their behavior accordingly.

We have elaborated on this issue in the Discussion (pages 19-20), highlighting the significance of our settings and acknowledging that this is only one of the possible interactive scenarios that might be explored in future research.

Comment#2.8

I had trouble understanding which statistical analyses were conducted for which research question: At the beginning, the authors say they were using mixed-effect logistic models, whereas at the end, they said they used nonparametric Tests such as Wilcoxon throughout the paper. Maybe a specification such as: to analyze research question a, we used method b, to analyze question x, we used method y would be helpful here.

Response#2.8

We thank reviewer #2 for the suggestion. We completely restructured and extended the "Statistical data analysis" section, now called "Research questions and statistical data analysis", differentiating between research questions and types of variables. The distinction based on research questions recall the research questions and relative hypotheses introduced in the Introduction section, following comment #2.5 of the same reviewer.

Comment#2.9

Results:

The first graph should be presented earlier for a better illustration of the written results.

Response#2.9

We thank the reviewer for the useful suggestion; we have moved Figure 1 to the Introduction section, where we introduce the experimental task (page 4).

Comment#2.10

In my personal opinion, the last paragraph on p. 12 would fit better in the discussion than in the results section, as it comprises a certain level of interpretation already (e.g., "took much more into consideration", "relied significantly less on the adult's advice"). As opposed to presenting the mere numeric differences as usual in the results section.

Response#2.10

We thank the reviewer for the suggestion; we have moved these considerations in the Discussion section (pages 17-18).

Comment#2.11

Just wondering: Wouldn't one label subsamples such as these of the different age groups as small "n" instead of capital "N"? (e.g., in Figure 3)

Response#2.11

We thank the reviewer for pointing this out: we have changed the labels using the small “n”.

Comment#2.12

Can you explain why condition wasn't used as an independent variable in a mixed model design? I would have expected that. Why do you incorporate condition into a difference score instead, using it as a dependent variable? I wonder why the authors wouldn't choose the former, more straightforward way?

Response#2.12

We are not sure to have understood to which analysis reviewer#2 is referring. All the mixed-effect models that tested differences in experimental conditions (i.e., Model 6, 7, 8) to investigate the emergence of reciprocity in participants' social influence do use condition as a (categorical) independent variable, as suggested by the reviewer. We are sorry if some of the formulations in the main manuscript and Supplementary Information were misleading. We have tried to clarify as much as possible this aspect in the revised versions of Manuscript and Supplementary Information.

Maybe the reviewer is referring to the first analysis of the “Reciprocity” section, the one testing the effect of age (as continuous variable) on reciprocity (mean influence in the Susceptible condition – mean influence in the Unsusceptible condition) through a multiple regression analysis (page 14). If this is the analysis mentioned by reviewer#2, by using the suggested approach we should have run a mixed-effects logistic model with final decision (binary choice) as dependent variable, condition and age (treated as continuous variable) and their interaction as independent variables. The problem with this analysis is that it tells something different from the analysis we have actually carried out and cannot specifically answer our research question about reciprocity. First, the main effect of condition in such a model reveals the estimated beta coefficient of the dependent variable at the intercept (i.e., when age is 0), which is rather meaningless. Then, we could look at the interaction effect of condition and (continuous) age. This effect tells something slightly different from the results of our main analysis on reciprocity. A significant interaction between condition and continuous age would reveal that age has a different impact on participants' final decisions in the two conditions. For instance, it would reveal that participants' willingness to follow the adult's estimate increases with age with a certain slope in the Susceptible condition, whereas it increases with a lower slope in the Unsusceptible condition. This effect might be interpreted, in principle, as an indirect signal of reciprocity, but it is a much less straightforward, direct and clear way to present an effect of condition in terms of final decisions (i.e. conformity to the adult's decision). Indeed, this interaction does not express a direct within-subject difference in the dependent variable across conditions (i.e. participants increase their susceptibility to the adult in the Susceptible condition), but tells us about the between-subject relationship between age and the dependent variable in each condition.

We did use the suggested approach when treating age as a categorical factor, since we could specifically test the effect of condition in each age cohort (Model 6, 7, 8) to specifically investigate whether reciprocity emerges at a specific stage of development. On the contrary, when analyzing the effect of age (as continuous variable) on the emergence of reciprocity, we used a multiple regression using as dependent variable the within-subject difference in conformity to the adult's estimate between Susceptible and Unsusceptible conditions, which is an individual measure of reciprocity. This measure has been used in Figure 3.b and has been used in recent works investigating direct and indirect reciprocity of social influence in adult-adult interaction through experiments with similar block designs (Mahmoodi et al., 2018 [73]; Zonca et al., 2021[74]).

Comment#2.13

Discussion:

On p. 17, the authors sum up a well-selected body of research on the cognitive abilities needed for decisions as demanded in the study. Could you please add the ages here? Because, what is missing in the discussion

is an explanation why 10-year-olds might show qualitatively different behavior than 6- and 8-years-olds. Do the authors have a suggestion for an underlying mechanism that is at work in the one, but not the other age groups?

Response#2.13

We thank the reviewer for the comment. We agree that adding information about the ages would clarify a bit more the potential contribution of specific cognitive skills for the emergence of social mechanisms such as conformity and reciprocity. We have expanded this section of the discussion in the revised version of the manuscript (pages 18-19), providing additional information on the cited literature and the relative age effects. Based on this literature, we have tried to provide hypotheses on the cognitive abilities that might play a role in the emergence of reciprocity in social influence contexts in 10-year-old children.

Comment(s)#2.14

Some minor comments/ typos:

p. 2, line 17: “3/4 years of age” reads like 0.75, maybe change to “3 to 4 years of age”

p. 7, l. 14: there is an “s” missing after participant

p. 9, l. 47: the “i” of influence should be in italics.

First line of the discussion: “behavioral” should be without the –al, or there is a noun missing

p. 16, l. 26: “children did reciprocated”, omit the “d”

I am not a native speaker, but based on the literature, “10 years old children” seems uncommon, wouldn’t one say “10-year-old children” instead, without the “s”?

Response(s)#2.14

We thank the reviewer for pointing out these typos. We have amended the revised manuscript accordingly.